# Bioactive Compounds from Medicinal Plants as Potential Adjuvants in the Treatment of Mild Acne Vulgaris

**DOI:** 10.3390/molecules29102394

**Published:** 2024-05-19

**Authors:** Mariateresa Cristani, Nicola Micale

**Affiliations:** Dipartimento di Scienze Chimiche, Biologiche, Farmaceutiche ed Ambientali, Università di Messina, Viale F. Stagno D’Alcontres 31, I-98166 Messina, Italy; mariateresa.cristani@unime.it

**Keywords:** mild acne vulgaris, bioactive compounds, plant extracts, phytochemicals, skin diseases, natural products, alternative medicine, adjuvants

## Abstract

In recent years, there has been a growing interest in the use of medicinal plants and phytochemicals as potential treatments for acne vulgaris. This condition, characterized by chronic inflammation, predominantly affects adolescents and young adults. Conventional treatment typically targets the key factors contributing to its development: the proliferation of *Cutibacterium acnes* and the associated inflammation. However, these treatments often involve the use of potent drugs. As a result, the exploration of herbal medicine as a complementary approach has emerged as a promising strategy. By harnessing the therapeutic properties of medicinal plants and phytochemicals, it may be possible to address acne vulgaris while minimizing the reliance on strong drugs. This approach not only offers potential benefits for individuals seeking alternative treatments but also underscores the importance of natural remedies of plant origin in dermatological care. The primary aim of this study was to assess the antimicrobial, antioxidant, and anti-inflammatory properties of plants and their phytochemical constituents in the management of mild acne vulgaris. A comprehensive search of scientific databases was conducted from 2018 to September 2023. The findings of this review suggest that medicinal plants and their phytochemical components hold promise as treatments for mild acne vulgaris. However, it is crucial to note that further research employing high-quality evidence and standardized methodologies is essential to substantiate their efficacy and safety profiles.

## 1. Introduction

Acne vulgaris (AV) is a common skin disease that mainly affects adolescents and young adults, particularly males. The causes of this disorder are strongly associated with particular factors, including bacterial colonization and inflammation. The main clinical manifestations of AV are non-inflammatory and inflammatory lesions, which occur primarily on the face, neck, trunk, and back [1]. This long-term skin condition is in most cases mild and not considered self-limiting. However, it can have a major impact on individuals’ quality of life and is often associated with the development of psychological disorders [2]. In addition to bacterial infection and an inflammatory state, AV is characterized by hyperplasia of sebaceous glands with overproduction of sebum and hyperkeratinization of the sebaceous ducts. Hormonal changes, especially during puberty, with the associated increase in androgen levels, are also considered triggers of the condition [3]. Bacterial colonization and proliferation, mainly by *Cutibacterium acnes* (formerly known as *Propionibacterium acnes*), an anaerobic Gram-positive commensal bacterium, stimulates inflammatory and immune responses. Virulence factors released by this bacterium include lipases, proteases, and hyaluronidases, as well as porphyrins, which can generate reactive oxygen species (ROS) and stimulate the production of chemokines and prostaglandins (PGs) by keratinocytes [4].

Multiple mechanisms have been proposed to elucidate how *C. acnes* exacerbates acne conditions, as illustrated in Figure 1. *C. acnes* contributes to comedogenesis by producing oxidized squalene and free fatty acids, resulting in a qualitative alteration in sebum. In addition, it activates the insulin-like growth factor (IGF)-1/IGF-1 receptor signaling pathway, increasing filaggrin expression, upregulating integrin-α3, -α6, and vβ6 levels, and influencing keratinocyte proliferation and differentiation, ultimately culminating in comedone formation. The pathogen can secrete metabolites that induce host-tissue degradation and inflammation, such as lipases, proteases, hyaluronate lyase, and cyclic adenosine monophosphate (CAMP) factors. Additionally, *C. acnes* can produce extracellular polymeric substances (EPS) and form biofilms, contributing to antibiotic resistance. *C. acnes* triggers and exacerbates inflammation by also activating Toll-like receptors (specifically TLR-2 and TLR-4) on keratinocytes, causing activation of mitogen-activated protein kinase (MAPK) and nuclear factor kappa-light-chain-enhancer of activated B cell (NF-κB) pathways. Subsequently, keratinocytes produce interleukins (IL)-1, IL-6, IL-8, tumor necrosis factor-α (TNF-α), granulocyte-macrophage colony-stimulating factor, human β-defensin-2 and matrix metalloproteinases. Additionally, the surface antigen cluster of differentiation 36 (CD36) recognizes *C. acnes*, inducing the production of ROS by keratinocytes, which eliminate bacteria and induce inflammation. Sebocytes are also involved in the inflammatory response; sebocytes TLR-2 recognize *C. acnes* and activate the NF-κB pathway, thus promoting inflammation. *C. acnes* can also be recognized by TLR-2 expressed on monocyte/macrophage lineage cells, causing production of the inflammatory cytokines IL-8 and IL-12. Additionally, *C. acnes* stimulates caspase-1 and NLRP3 inflammasome gene expression in monocytes, leading to an excess of IL-1 production. *C. acnes* also exhibits T cell mitogenic activity, with an induced adaptive immune response involving CD4+ T lymphocytes, particularly T helper (Th)1 and Th17 cells. This leads to the secretion of IL-6, IL-1, and transforming growth factor-β (TGF-β) from peripheral blood mononuclear cells, promoting the differentiation of naive CD4+CD45RA T lymphocytes into Th1 and Th17 cells. As a result, Th effector cytokines, such as IL-17 and interferon-γ, are upregulated [3,5].

According to European guidelines, topical agents are recommended for mild AV; treatment includes retinoids, benzoyl peroxide, azelaic acid, and antibiotics. Retinoids reduce sebum production and normalize epithelial desquamation, as well as having anti-inflammatory activity. Benzoyl peroxide has antibacterial and anti-inflammatory activities and exhibits mild comedolytic activity. Similarly, azelaic acid has antimicrobial, anti-inflammatory, and comedolytic properties and does not give rise to bacterial resistance [6].

Although several therapeutic options exist for topical treatment of mild acne, side effects, insufficient response to therapy, and high costs of some drugs are prompting the scientific community to seek alternative and complementary therapies, particularly those of natural origin [7]. Considering the need to develop alternative therapies for the treatment of AV and also taking into account the mechanisms by which the bacterium carries out its action, numerous studies that have successfully tested medicinal plants and phytochemicals in the treatment of mild AV have emerged; this motivated the writing of this review. In particular, this study focused on reviewing phytotherapy studies conducted in the past 5 years that have significant mild anti-acne potential. The main features of the studies included in this review article are summarized in Table 1 (see hereinafter).

## 2. Methodology

### Search Strategy and Inclusion and Exclusion Criteria

Three electronic databases (PubMed, Web of Science, and Scopus) were searched from 2018 to September 2023.

The search strategy used for PubMed functioned as an indication for the search strategies in other databases;The research included the term “acne vulgaris” combined with the terms “plants”, “extracts”, “clinical trial” using boolean operator tools AND, OR, NOT; studies did not include clinical trials;Overall, 89 articles were found as a result of the search;A total of 35 studies were considered relevant by us and therefore included in this review;In these 35 selected studies, the efficacy of herbal medicine in the treatment of AV was evaluated by considering in vitro and ex vivo experiments.

In this review, we discussed and reported the main bioactive compounds from natural sources that have shown significant beneficial effects in the treatment of mild AV, highlighting potential mechanisms of action where identified. Studies have mainly considered extracts from natural products and, on some occasions, essential oils (EOs). Their biological activities were tested on different cell lines, mainly keratinocytes, fibroblasts, monocytes, and sebocytes; moreover, cytotoxicity was evaluated to demonstrate the safety of the samples. The antimicrobial activity has often been tested not only on *C. acnes* but also on other bacterial strains such as *S. aureus* and *S. epidermidis* that may significantly contribute to the pathogenesis of AV [43].

Considering the results of these studies and the reliability of the experimental evidence presented, we can state that the botanical species with anti-acne medicinal properties are varied and with different characteristics. The most active natural component is characterized in most cases by a phenolic structure responsible for antioxidant action. However, other molecules have also shown efficacy in anti-acne treatment, especially considering the effects (for example, the anti-lipase, anti-tyrosinase, and anti-inflammatory effects) associated with the virulence factors released by the pathogen [4].

Through this review, it was possible to ascertain that a total of 40 medicinal plant species belonging to 25 families (mostly *Lamiaceae*, *Anacardiaceae*, *Poaceae*, and *Rosaceae*, but also *Apiaceae*, *Asteraceae*, *Brassicaceae*, *Cannabaceae*, *Caprifoliaceae*, *Celastraceae*, *Cistaceae*, *Fagaceae*, *Hamamelidaceae*, *Lauraceae*, *Meliaceae*, *Myrtaceae*, *Musaceae*, *Papaveraceae*, *Rubiaceae*, *Salicaceae*, *Sapindaceae*, *Smilacaceae*, *Theaceae*, *Urticaceae*, and *Zingiberaceae*) have demonstrated an important role in the treatment of AV due to their multiple biological properties.

## 3. Families and Study

### 3.1. Lamiaceae

The *Lamiaceae* are one of the largest families of flowering plants comprising approximately 250 genera and over 7000 species. Most of the plants in this family are aromatic and are therefore sources of EOs. These plants are widely used as cooking herbs and as medicinal plants in various folk traditions. In the Mediterranean area, oregano, sage, rosemary, thyme, and lavender stand out for their geographical spread and variety of uses [44]. Several widely distributed plants are also part of this family, including the following: *Callicarpa americana* L., a shrub native to the American southeast used in traditional medicine, characterized by the presence of biologically active terpenoids [12], and *Plectranthus madagascariensis* (Pers.) Benth., a species of aromatic plant indigenous to South Africa traditionally used for the treatment of various dermatological and respiratory ailments. The most interesting varieties are *Plectranthus aliciae* (Codd) van Jaarsv. & T.J. Edwards; *Plectranthus ramosior* (Benth.) van Jaarsv.; *Plectranthus madagascariensis* (Pers.) Benth var. *madagascariensis*, rich in iridoids and iridoid glycosides, phenylpropanoid glycosides, organic acids, volatile oils, terpenoids, saccharides, flavonoids, sterols, and saponins [13,45]; and *Scutellaria baicalensis* Georgi, which in China still plays an important role in traditional medicine with functions of clearing away heat and dampness, purging fire and detoxification. This medicinal plant rich in flavonoids, terpenoids, volatile oils, and polysaccharides is widely distributed in Russia, Mongolia, North Korea, and Japan, as well as in China [46].

Chuang L.-T. et al. (2018) demonstrated that ethanolic extracts of *Origanum vulgare* L. leaves rich in phenolic compounds exert anti-inflammatory effects in human THP-1 monocytes by suppressing production and over-expression of pro-inflammatory IL-8, IL-1β, and TNF-α, in part by blocking the TLR2-mediated NF-κB signaling pathway. Moreover, these extracts significantly suppressed *C. acnes*-induced skin inflammation (expressed as ear thickness and biopsy weight) in in vivo mouse ear edema models after intradermal injection. Four major compounds were identified as responsible for the aforementioned biological activities, namely rosmarinic acid, quercetin, apigenin, and carvacrol (Figure 2) [8].

Taleb M.H. et al. (2018) assessed the potential anti-acne activity of selected EOs obtained from plants used in Mediterranean folk medicine by testing their antimicrobial activity against *C. acnes* and *S. epidermidis*. The most effective EOs were obtained from plants of the *Lamiaceae* family, i.e., oregano and thyme (*Thymus vulgaris* L.). In both species, the most active and abundant component was identified in the monoterpenoid phenol derivative thymol which showed low minimum inhibitory concentration (MIC) (0.70 mg/mL) and minimum bactericidal concentration (MBC) (1.40–2.80 mg/mL) values against both bacterial strains. An oregano EO was also formulated and tested as a topical nanoemulsion in an in vivo acne murine model showing higher efficacy than the reference antibiotic (erythromycin) [9]. An equatable study was carry out more recently by Abdelhamed F.M. et al. (2022), in which five EOs (extracted specifically from tea tree, clove, thyme, mentha, and basil) were taken into analysis and tested for their antimicrobial activities in vitro and, as topical nanoemulsions, for anti-inflammatory properties in vivo, using the same bacterial strains and mouse model. In this study, the thyme EO, rich in (poly)phenols and terpenoids, proved to be the most effective extract. The in vitro studies displayed its remarkable antimicrobial and antibiofilm activity. Furthermore, thyme EOs have been shown to affect cell membrane permeability/integrity (leakage of K^+^ ions and nucleic acids) and induce morphological changes in both bacterial species. The in vivo experiments evidenced that the thyme EO nanoemulsion suppresses the inflammatory response in acne mouse models, decreasing bacterial load and healing ear skin [10].

Oliveira A.S. et al. (2022) studied *Thymus* × *citriodorus* (Pers.) Schreb. (TC) (also lemon thyme or citrus thyme), an interspecific hybrid between *Thymus pulegioides* L. and *Thymus vulgaris* L. known for its use in folk medicine as a seboregulator with anti-acne effects. In detail, the authors evaluated the anti-acne potential of two TC preparations for topical applications, an EO and a hydrolate. The main bioactive compounds present in the EO were the monoterpenoids geraniol, followed by eucalyptol (1,8-cineole) and thymol; for the hydrolate, the main constituents were also monoterpenoids, namely eucalyptol, followed by linalool and geraniol (Figure 3). The EO showed direct antimicrobial activity for *C. acnes* and *S. epidermidis*, whereas the hydrolate revealed visual MIC only for *C. acnes.* The EO was also effective in preventing biofilm formation and disrupting preformed biofilms even at sub-inhibitory concentrations. In contrast, the TC hydrolate showed modest anti-biofilm effects. Regarding the anti-inflammatory activity profile, both TC preparations were able to inhibit nitric oxide (NO) production at non-cytotoxic concentrations in lipopolysaccharide (LPS)-stimulated mouse macrophages, while showing no NO or ROS scavenging capacity. The hydrolate also showed superior biocompatibility compared to the EO assessed using a *Daphnia magna* acute toxicity assay [11].

Pineau R.M. et al. (2019) showed that *Callicarpa americana* L. ethanol leaf extracts and EOs exhibit antimicrobial, anti-inflammatory, and antioxidant activity. The evaluation of the antiproliferative activity of fractions of the extract isolated by flash chromatography (LC-FTMS ESI-positive mode analysis) against a panel of 10 distinct *C. acnes* isolates brought about MIC and IC_50_ values in the range 16–32 μg/mL and 4–32 μg/mL, respectively. Notably, the same fractions displayed a remarkable selectivity index (up to >128) evaluated against human adult keratinocytes (HaCaT cells). In these extracts were identified flavonoids such as genkwanin, 5-hydroxy-7,4′-dimethoxyflavone (i.e., genkwanin 4′-methyl ether), and luteolin (Figure 4). The leaf EO contained lipids [e.g., (*E*)-2-hexenal and 1-octen-3-ol], monoterpenoids (i.e., nopinone, α-pinene, and β-pinene), sesquiterpenoids (i.e., α-cadinol, caryophyllene oxide, 7-epi-α-eudesmol, α-humulene, humulene epoxide II, intermedeol, khusinol, valencene, α-selinene, and 7-epi-α-selinene), and triterpenoids (e.g., euscaphic acid) (Figure 5) [12].

Zhu X. et al. (2020) proved that the flavonoid wogonin present in *Scutellaria baicalensis* Georgi extracts exerts better anti-acne effects than its glycoside wogonoside (wogonin 7-glucuronide) (Figure 6) in inhibiting the up-regulation of IL-1β and IL-8 levels caused by *C. acnes* through inactivation of the MAPK and NF-κB signaling pathways. Next, they devised an eco-friendly strategy, namely fermentation of plant extracts by the symbiotic fungus *Penicillium decumbens* f3-1, to convert the glycoside into its aglycone (conversion rate = 91% in 4 days). In contrast, no significant anti-acne potential (IL-1β inhibition rate < 50%) was detected for the second most abundant flavonoid in the extracts and its glycoside, namely baicalein and baicalin [14].

### 3.2. Anacardiaceae

The *Anacardiaceae* are a plant family (also known as the cashew family) that includes several species of high economic importance from a nutritional/food standpoint (the best-known being cashew, pistachio, mango, and pink pepper), as well as plants of local importance for traditional medicine. Members of this family are known to be particularly rich in polyphenols [47]. Mango (*Mangifera indica* L.) is widely cultivated in tropical regions, especially in India and Thailand, where it has been an important medicinal plant in traditional ayurvedic and indigenous medicine for more than 4000 years. Its parts (especially the leaves) are still commonly used as remedies to treat burns, scalds, and related infections due to its high content of polyphenols [16]. Cashew (*Anacardium occidentale* L.) is a tropical plant native to South America (especially widespread in the northeastern region of Brazil) rich in tannins, carotenoids, and polyphenols [48]. Extracts from both of these products have recently been studied for possible beneficial effects in the treatment of AV.

Poomanee W. et al. (2018) showed that extracts derived from raw and ripe fruit kernels of *M. indica* L. grown in northern Thailand (Kaew-Moragot cultivar) exhibit significant antimicrobial activities against *C. acnes* and *Staphylococci* (*S. aureus* and *S. epidermidis*), especially the ethanolic fractions and the crude ethanol extracts (MIC values 1.56–3.13 mg/mL). Additionally, these extracts demonstrated potent antioxidant activity and displayed anti-inflammatory properties by inhibiting the secretion of IL-8 in LPS-stimulated RAW 264.7 cells [15].

De Tollenaere M. et al. (2022) demonstrated that leaf extracts at 0.3% *M. indica* L. have a positive impact on skin microbiota equilibrium and the seboregulation of sebocytes. Specifically, in vitro lipogenesis inhibition assays performed on human SEBO662AR sebocyte cell lines displayed a 40% reduction in lipid content after 7 days of treatment with the extract. A lower reduction (22%) was detected using mangiferin (Figure 7), the main phytomarker identified, indicating that other phytochemicals are also involved in the seboregulation. Clinical investigations were conducted in Caucasian volunteers with AV who received a treatment a cream containing *M. indica* extracts at 1% for 28 days, which showed very positive outcomes in terms of qualitative/quantitative sebum production and porphyrin expression. Docking studies were performed with iriflophenone and maclurin, two aglycones of the identified secondary phytomarkers (the fourth was penta-*O*-galloyl-β-D-glucose; Figure 7), revealing that both compounds bind effectively to the peroxisome proliferator-activated receptor (PPAR)γ transcription factor involved in modulating lipogenesis. Ex vivo experiments (extract at 1%) showed a significant reduction in the production of squalene (−18%), free fatty acids (−8%), and porphyrins [16].

Cefali L.C. et al. (2020) exploited the recovery of by-products of the food industry as a green strategy to obtain formulations with high anti-acne and anti-aging potential. Specifically, they used peduncle (pulp) extracts of cashew *(Anacardium occidentale* L.) with a high content of flavonoids (mainly rutin). In vitro experiments showed that these extracts, despite the poor antibacterial activity profile against *C. acnes*, were endowed with skin healing and antioxidant properties without any cytotoxic effects on keratinocytes. The extracts were also incorporated into an oil-in-water emulsion, resulting in a promising topical formulation for use as an anti-acne treatment [17].

### 3.3. Cannabaceae

The *Cannabaceae* family encompasses approximately 117 species found across tropical, subtropical, and temperate regions worldwide. These members are predominantly trees or shrubs, with occasional occurrences of vines (genus *Humulus*) or erect grasses (genus *Cannabis*), showing significant diversity in both morphology and habitat. Economically important plants in this family include marijuana or hemp (*Cannabis sativa* L.) and hops (*Humulus lupulus* L.). Hemp, believed to be one of the earliest domesticated plants, probably dates back to early Neolithic times in China and has been a vital source of fiber, food, and medicine for millennia. Female inflorescences of hops have been an integral part of brewing since the early Middle Ages. Another economically notable species is the wingceltis (*Pteroceltis tatarinowii* Maxim.), prized for its bark fiber, which serves as the primary material in the production of traditional Chinese Xuan paper [49].

Jin S. et al. (2018) showed that *Cannabis sativa* L. seed hexane extracts, besides the direct bactericidal effect on *C. acnes*, are able to suppress expression of inflammatory enzymes such as inducible NO synthase (iNOS) and cyclooxygenase (COX)-2 (assessed by Western blot analysis performed on *C. acnes*-infected HaCaT cells) and reduce the levels of the related products NO and PGE_2_. The expression of the pro-inflammatory cytokines IL-1β and IL-8 was also reduced after treating the infected cells with the extracts. These extracts, which contain high levels of polyunsaturated fatty acids including linoleic acid, oleic acid, *cis*-11-eicosenoic acid, palmitic acid, γ-linolenic acid, arachidic acid, palmitoleic acid, and heneicosanoid acid, turned out to also be effective in reducing the lipogenesis in IGF-1-induced sebocytes via regulating the AMPK and AKT/FoxO1 signaling pathways and inhibiting 5-lipoxygenase (5-LOX) activity. They also promoted collagen biosynthesis in vitro by inhibiting the gelatinase MMP-9 in *C. acnes*-infected Hs68 cells [18].

Weber N. et al. (2019) studied the antibacterial, antioxidant and anti-inflammatory properties of a hop-CO_2_ extract derived from the flowers of *Humulus lupulus* L. on human primary keratinocytes [19]. The antibacterial activity of this hop extract was assessed against four strains of *C. acnes* and four strains of *S. aureus*, providing noteworthy MIC values of 3.1 µg/mL and 9.4 µg/mL, respectively. Superior antibacterial effects were obtained with a gel formulation (0.3% hop extract *w*/*w*). The hop extract, rich in humulnones (humulone, adhumulone, and cohumulone) and lupulones (*n*-lupulone, adlupulone, and colupulone) (Figure 8), was standardized to ~50% content of these bioactive components by adding sunflower oil before biological assessments. Surprisingly, this extract did not contain xanthohumol, a compound that has been shown to possess strong inhibitory activity against *C. acnes* in previous studies [50]. The bactericidal properties of these acidic derivatives have been attributed mainly to the presence of the isoprenoid side-chains which enable disruption of the bacterial membrane integrity, causing leakage and inhibiting the transport of nutrients [51]. These hop compounds are also endowed with proton ionophore activity and strong redox reactivity, resulting in oxidative damage to cellular structures. Some of them, especially lupulone and xanthohumulone, have the capability to penetrate biofilms formed by *Staphylococcus* species, including methicillin-resistant strains, and reduce bacterial populations within these biofilms. The extract also showed good antioxidant activity (IC_50_ = 29.43 µg/mL) and anti-inflammatory properties (IC_50_ = 0.8 µg/mL) by reducing IL-6 expression.

### 3.4. Poaceae

The *Poaceae* (or *Gramineae*) are the most economically important plant family because of their fundamental role as the main source of food for humans (staple) and animals (grasses). In general, these plants contain a wide array of chemical classes of compounds endowed with biological activity. Although several phytochemicals derived from plants of this family have been isolated and their therapeutic benefits proven, the pharmacological and cytotoxic profile of the *Poaceae* still remain uncertain if compared to other plant families due to limited scientific evidence [52].

Kim C. et al. (2022) conducted a comparative study on the anti-AV potential of lemongrass (*Cymbopogon citratus* Stapf.) extracts. From this study, it emerged that the ethyl acetate extract has superior ROS- and NO-scavenging properties compared to 80% methanol, *n*-hexane, *n*-butanol, and water extracts. This extract also showed the highest elastase and collagenase inhibitory activity which makes it suitable for cosmeceutical applications (anti-aging properties). Its tyrosinase inhibitory activity (whitening properties) instead was comparable to those of the other extracts. The ethyl acetate fraction displayed higher lipase and antimicrobial (*C. acnes*) activities among extracts in line with its higher content of phenolic acids (cinnamic acid, caffeic acid, salicylic acid, *p*-hydroxybenzoic acid, gallic acid, ferulic acid, and protocatechuic acid) and flavonoids (isovitexin, luteolin, catechin, tricin, and chrysoeriol 7-*O*-glucoside) (Figure 9) [20].

Rodríguez-López J. et al. (2022) highlighted the enormous potential in anti-AV topical formulations of a biosurfactant extract obtained from corn (*Zea mays* L.) steep water, a by-product of the milling industry. This biosurfactant extract (0–5%) has been employed as an ingredient in formulations along with antimicrobial ZnO (0–2%) and anti-inflammatory salicylic acid (0–2%). In this study, a clear synergistic antimicrobial effect was detected between ZnO and the biosurfactant extract in the absence of salicylic acid. Interestingly, in addition to its own antimicrobial activity, the biosurfactant extract in formulations with an intermediate concentration of ZnO (1%) showed inhibitory activity against *C. acnes* that was higher than ZnO alone and similar to that of formulations containing ZnO (1%) and salicylic acid (1%) [21].

### 3.5. Rosaceae

The *Rosaceae* are a family (known as the rose family) of medium-sized flowering plants comprising more than one hundred genera and more than 3000 recognized species, including fruit, nut, ornamental, aromatic, herbaceous, and woody plants. Numerous bioactive compounds that offer important health benefits have been identified in these plants. Therefore, the study of the phytochemical composition of these species, particularly the less economically used *Rosaceae*, may prove to be of crucial importance for drug development [53]. The medicinal use of these plants is widely described in the scientific literature. The genus *Cotoneaster*, one of the most representative of the *Rosaceae* family, encompasses about 500 species with distribution and habitat in the Eurasian region. Their greatest biodiversity is found in the mountains of China and the Himalayas, where these plants have taken on a primary role in traditional medicine. This genus, which includes mainly shrubs and small trees, provides ornamental plants used for landscaping due to their diversity of forms, glossy green leaves, abundant flowers, and attractive fruits [54].

Krzemińska B. et al. (2022) studied plants of this genus in depth, highlighting their healing potential in skin diseases. In their first work, they evaluated the chemical composition and biological activity profile of *Cotoneaster nebrodensis* (Guss.) K. Koch and *Cotoneaster roseus* Collett extracts (from fruits and leaves). In vitro cell-based experiments have shown that both plant extracts possess notable antimicrobial activity without any toxicity towards skin fibroblasts. The antimicrobial potential of these extracts also depended on the solvent (or mixture of solvents) used for the extraction. These extracts also showed significant antioxidant and anti-inflammatory properties, the latter assessed as lipoxygenase, hyaluronidase, COX-1, and COX-2 inhibition. The most abundant flavonoids present in both species were quercetin derivatives (in order of quantity: quercitrin, astragalin, and isoquercitrin). Rare flavonoids (sissotrin and 5-methyl-genistein-4′-*O*-glucoside) were also detected in significant amounts (Figure 10). Among phenolic acid derivatives, apart from the most abundant and common chlorogenic acid, atypical compounds in leaf extracts were the aromatic esters cotonoate A and horizontoate A, and the sphingolipid horizontoate C. Scopoletin was the most abundant coumarin derivative found in these species (Figure 11) [23]. In a second work conducted in parallel, the authors studied two other species of the genus *Cotoneaster*, namely *C. hsingshangensis* J.Fryer & B.Hylmö and *C. hissaricus* Pojatk for which they came up with similar results [22].

### 3.6. Asteraceae

The *Asteraceae* (formerly known as *Compositae*) are a family of flowering plants of great economic importance, providing food staples, garden plants and herbal medicines, characterized by the presence of numerous clustered inflorescences, which have the appearance of a single compound flower. It is estimated that this family accounts for about 10% of all flowering species. It is also considered one of the most evolved and biodiverse plant families (~32,000 species) among the dicotyledons, found in almost every environment on the planet except Antarctica [55]. Miazga-Karska M. et al. highlighted the anti-bacterial, anti-biofilm and antioxidant effects of low molecular weight (<5000 Da) peptides isolated from burdock (*Arctium lappa* L.) roots. These peptides were shown to be active against two strains of *C. acnes* and two strains of *Staphylococcus* (*S. aureus* and *S. epidermidis*) without cytotoxic effects on human fibroblasts (SI = 160–320). Their antioxidant potential was found to be much higher in acne treatment dressing materials obtained by cross-linking polysaccharides (chitosan and alginate) with the aglycone genipin. The antioxidant mechanism of these biomolecules can be exerted through free radical scavenging, chelation of pro-oxidative transition metals, and reduction of hydroperoxides) [24].

### 3.7. Caprifoliaceae

The Caprifoliaceae family (or honeysuckle family) is represented by approximately 960 species, most of which are distributed in temperate regions of the northern hemisphere. This family is widely used in traditional Chinese medicine as well as in Japanese medicine to cure febrile illnesses, upper respiratory tract infections, sores, swellings, and pneumonia [56].

Chrząszcz M. et al. conducted an analytical and comparative study between extracts of two plants of the genus *Cephalaria*, namely *C. uralensis* (Murray)Roem. & Schult and *C. gigantea* (Ledeb.) Bobrov. Analysis of the extracts revealed the presence of about 40 bioactive compounds, including chlorogenic acid (see Figure 11), swertiajaponin (a well-known skin-whitening flavonoid derivative which inhibits both activity and protein expression levels of tyrosinase) [57], and isoorientin (the 6-*C*-glucoside of luteolin) (Figure 12), those most present and to which the antioxidant, anti-inflammatory, antibacterial, and anti-acne activities are presumably attributed. In particular, the ethanolic extract of the aerial parts of *C. uralensis* showed the best biological activity profile (i.e., substantial inhibition of COX-1 and COX-2, good radical scavenging properties, and, most importantly, no toxicity on normal skin fibroblasts), although the antibacterial activity against all tested bacterial strains tested (*S. aureus*, *S. epidermidis*, and *C. acnes*) was moderate [25].

### 3.8. Cistaceae

The *Cistaceae* are a relatively small plant family (also known as the rock-rose or rock rose family) with beautiful shrubs covered by flowers at the time of blossom. This family consists of about 170–200 species distributed mainly in the temperate areas of Europe and the Mediterranean basin, also found in North America; a limited number of species are found in South America. The genus *Cistus* includes fragrant species with a high content of flavonoids and bioactive terpenes, which are widely used in folk medicine for the treatment of various pathologic conditions [58]. Bouabidi M. et al. investigated the bioactivity of extracts of *C. laurifolius* L. and *C. salviifolius* L. Analysis of the chemical composition of these extracts revealed the presence of several polyphenols (especially flavonoids such as myricetin, quercetin, and kaempferol; Figure 13) and ellagitannins (mainly terflavin A and cistusin; Figure 13). The extracts also showed a good antimicrobial profile, particularly against *S. aureus*, *S. epidermidis*, and *C. acnes* [26].

### 3.9. Fagaceae

The *Fagaceae* family includes eight genera and approximately 927 species of flowering plants. The leaf features of *Fagaceae* can closely resemble those of *Rosaceae*. The *Fagaceae* stand out as a crucial family of woody plants in the northern hemisphere, playing a pivotal role in the temperate forests of North America, Europe, and Asia, particularly due to oaks, which serve as a vital food source for wildlife. The genus *Quercus* is the one mainly associated with traditional medicine to treat and prevent various human disorders ranging from asthma to gastrointestinal diseases [59].

Kim M. et al. elucidated the anti-AV potential of a *Quercus mongolica* Fisch. leaf extract and its primary bioactive compound, namely the ellagitannin pedunculagin (Figure 14). The medicinal potential of the *Q. mongolica* Fisch. leaf extract is also related to its high content of flavonoids, tannins, triterpenoids, and phenols, which are known for their antioxidative, anti-inflammatory, antitumor, antimicrobial, and antiallergic properties. However, in this study, the authors demonstrated that both the leaf extract and the pure bioactive component exhibit significant anti-inflammatory activity, related to the inhibition of NO production and reduction in the levels of inflammatory cytokines such as IL-6 and IL-8. Furthermore, both *Q. mongolica* Fisch. leaf extract and pedunculagin displayed potent inhibitory activity against 5α-reductase type 1, the isoform of 5α-reductase most related to sebum production. No toxicity was detected on RAW 264.7 macrophages and HaCaT cells [27].

### 3.10. Hamamelidaceae

The witch hazel family, *Hamamelidaceae*, comprises approximately 30 genera and 140 species distributed mainly in subtropical and temperate regions. While their distributions are typically restricted, the genera *Hamamelis* and *Liquidambar* exhibit a disjunct intercontinental distribution in the north temperate zone [60].

In 2022, Piazza S. et al. assessed the biological activity profile of the glycolic extract of witch hazel (*Hamamelis virginiana* L.) bark against *C. acnes*-induced inflammation. Phytochemical analysis of the extract revealed that hamamelitannin and oligomeric proanthocyanidins were the most abundant compounds (Figure 15). This extract showed inhibition of *C. acnes*-induced IL-6 release, partially impairing NF-κB activation, although it lacked antibacterial and antibiofilm activity. In addition, this extract showed greater anti-inflammatory activity related to the inhibition of IL-8 release than hamamelitannin, evidently due to its high proanthocyanidin content and partially mediated by antioxidant mechanisms [28].

### 3.11. Lauraceae

*Lauraceae* (or the laurels) are a family of plants with worldwide distribution, found abundantly in tropical and subtropical regions. Due to its unique ecosystem, Taiwan is a country particularly rich in plants of this family (which form the renowned forests of Taiwan), where they have played a significant role in economics and folk medicine since ancient times. The phytochemicals of the *Lauraceae* are quite numerous and varied, as is their range of bioactivities, including anti-tubercular, anti-inflammatory, cytotoxic, and antiplatelet properties, which are attracting the attention of pharmaceutical research [61].

In a recent study, Yang C.L. et al. (2020) isolated for the first time from the stem of *Cinnamomum validinerve* a new dibenzocycloheptene derivative and a butanolide derivative (named validinol and validinolide, respectively; Figure 16) alongside 17 other known compounds. Among the isolates, three compounds, namely isophilippinolide A, secosubamolide, and cinnamtannin B1 (Figure 16), were reported as effective in vitro against *C. acnes* with MIC values in the range of 16–500 µg/mL, whereas lincomolide A, secosubamolide, and cinnamtannin B1 (Figure 16) exhibited potent inhibition of superoxide anion generation (IC_50_ range = 2.20–4.37 µM) and elastase release (IC_50_ range = 3.04–4.64 µM) by human neutrophils. Furthermore, cinnamtannin B1 (the major component) exerted anti-inflammatory properties after intraperitoneal injection in an in vivo ear *C. acnes*-infected murine model by reducing the levels of pro-inflammatory cytokines (TNF-α and IL-6) and immune cell infiltration [29].

### 3.12. Meliaceae

The *Meliaceae* family, commonly known as the mahogany family (Swietenia Jacq.), is renowned for its diverse applications, including cosmetics, medicinal use (antifungal, antiviral, and antibacterial), and even as a source of poisons such as insecticides. With approximately 48 genera and 700 species, *Meliaceae* are widely distributed throughout tropical regions and are also found in some temperate areas [62].

Kola-Mustapha A.T. et al. in 2023 carried out a computational study on bioactive compounds present in neem oil in order to speculate on their possible mechanisms of action for anti-acne activity. First, they performed extraction from neem (*Azadirachta indica* A. Juss.) leaves using a steam distillation method. Then, they examined the phytochemical components of the extract by GC-MS. Ten compounds met the drug-likeness requirements (Lipinski’s rule) and therefore underwent molecular docking on four selected targets for the treatment of AV. From this integrated study, it emerged that three of the main components of the neem oil, namely (2-(1-adamantyl)-*N*-methylacetamide), (*N*-benzyl-2-(2-methyl-5-phenyl-3*H*-1,3,4-thiadiazol-2-yl)acetamide) and (*N*-(3-methoxyphenyl)-2-(1-phenyltetrazol-5-yl)sulfanylpropanamide) (PubChem ID_610088, PubChem ID_600826, and PubChem ID_16451547 (Figure 17), respectively) possess high binding affinity towards the genes STAT1, CSK, CRABP2, and SYK [30].

### 3.13. Musaceae

The *Musaceae* are a family of flowering plants that holds significant economic importance comprising approximately 90 species. Originating from the hot, tropical regions of southeast Asia, *Musaceae* plants have also spread widely throughout the tropical areas of Africa. The largest genus of this family is *Musa*, to which bananas and plane trees belong. Cultivated bananas, including *Musa acuminata* Colla and *Musa balbisiana* Colla, are significant commercial members of the family, while many others are cultivated for ornamental purposes [63].

As the antimicrobial and anti-inflammatory properties of banana peels have been exhaustively assessed in previous studies, Savitri D. et al. determined to explore the skin protective effects of banana peel extracts from *M. balbisiana* in the case of AV. This study was conducted on an in vivo murine model and underlined remarkable antimicrobial and anti-inflammatory properties (via suppression of pro-inflammatory cytokine production, IL-1α, IFN-γ, IL-8, and TNF-α) of *M. balbisiana* extracts. Significant phytochemicals identified in these extracts were trigonelline, salsolinol, vanillin, isovanillic acid, ferulic acid, and rutin (Figure 18). Molecular docking revealed that the latter (rutin) possesses the highest binding affinity towards both TLR2 and NF-κB [31].

### 3.14. Papaveraceae

The family *Papaveraceae* (known as the poppy family) is also of considerable economic importance, boasting some 42 genera and approximately 775 known species of flowering plants within the order Ranunculales. This family is widely distributed in temperate and subtropical regions, predominantly in the northern hemisphere, encompassing areas such as East Asia and California in North America, with minimal occurrence in tropical regions. Although most members of this family are herbaceous plants, there are also shrubs and small trees. In particular, the *Papaveraceae* family is renowned for its isoquinoline alkaloids, such as berberine, tetrahydroberberine, protopine, and benzophenanthridine in the *Papaveroideae* sub-family, and spirobenzylisoquinoline and cularine in the *Fumarioideae* sub-family. These alkaloids, together with aporphine, morphinan, pavine, isopavine, narceine, and rhoeadine, contribute to the pharmacological properties of the plants in the family. The characteristic compounds of *Papaveraceae* also include meconic acid and chelidonic acid, as well as cyanogenic glycoside compounds derived from tyrosine, such as dhurrin and triglochinin, which are mainly found in the *Fumarioideae* sub-family. The *Chelidonieae* tribe contains the free amino acid δ-acetylornithine. In addition, flavonols such as kaempferol and/or quercetin are commonly found in these plants [64].

*Meconopsis quintuplinervia* Regel is a perennial herb utilized in traditional Tibetan folk medicine across China. It is employed to treat a wide range of conditions, such as headache, hepatitis, pneumonia, and edema. This plant grows mainly in Qinghai, Shanxi, Tibet, Gansu, and other regions of China. It was recently studied by Xie M. et al. (2023) regarding its potential for the treatment of AV. They evaluated the antibacterial properties of a *M. quintuplinervia* Regel extract against *C. acnes* and *S. aureus*. The study also entailed analysis of cell morphology, cell membrane/wall integrity, and protein and biofilm production. The extract contained various bioactive components, including alkaloids, flavonoids, and volatile oils. Of particular note were the antibacterial substances quercetin and luteolin, which are known to possess broad-range antibacterial activity (e.g., *E. coli*, *B. subtilis*, *S. typhi*, and *E. faecalis*). The findings of this study indicated that the extract exhibits significant antibacterial activity against both *P. acnes* and *S. aureus* and induces notable morphological changes in bacterial cells. Furthermore, leakage of alkaline phosphatase and nucleic acids confirmed that the bactericidal mechanisms occurred via disruption of the integrity of the bacterial membrane. Protein analysis revealed that the extract inhibits total protein expression and reduces adenosine triphosphatase activity. The bacterial biofilm production was also significantly suppressed and the adhesive capacity of bacterial cells compromised [32].

### 3.15. Rubiaceae

*Rubiaceae*, commonly known as the madder family and classified under the order Gentianales, encompasses about 13,500 species in 619 genera of herbs, shrubs, and trees. Its distribution is predominantly in tropical regions across the globe. Many species within this family hold economic significance due to their provision of valuable phytochemicals, while others are cultivated for their ornamental value. Extracts obtained from plants of this family have been used to treat various high-impact non-communicable diseases such as cancer, diabetes mellitus, acute hypertension, ischemia, and liver diseases. They may also cure asthma, cough, fever, gastric hyperacidity, jaundice, and peripheral edemas [65].

By exploiting a network pharmacological method, Seo G. and Kim K. investigated the anti-AV potential of bioactive compounds contained in *Hedyotis diffusa* Willd. In particular, they identified seven hit compounds (Figure 19) that meet the drug-likeness requirements using the Traditional Chinese Medicine Systems Pharmacology database. Molecular targets were collected from using the Swiss Target Prediction platform to perform comparative docking analyses (the anti-acne isotretinoin and standard antibiotics were used as reference compounds). From this study, it emerged that the mechanisms of action of these compounds mainly involve the regulation of lipid metabolism; to follow, adhesion of inflammatory cells, migration, ROS- and NO-production, and apoptosis. Overall, they hypothesized that *H. diffusa* Willd may exert anti-acne effects by directly or indirectly suppressing sebum secretion and inflammation [33].

### 3.16. Salicaceae

*Salicaceae* comprises approximately 650 plant species worldwide, classified into three genera: *Chosenia* Nakai, *Populus* L., and *Salix* L. Members of this family are known for their rapid growth and are mainly used for various economic purposes such as timber production, papermaking, fencing, shelter construction, snowshoe crafting, arrow shafts, fish traps, whistles, nets, and rope. They also serve as a source of biomass fuel, contributing to renewable energy, and find applications in ornamental, architectural, and horticultural landscaping. Additionally, they play a significant role in environmental conservation by aiding in soil erosion control. The genus *Salix*, commonly known as “the willow”, encompasses 330–500 species and over 200 hybrids. These species are predominantly distributed in the northern hemisphere, although a limited number are also found in the southern hemisphere. Plants of the genus *Salix* are widespread in Africa, North America, Europe, and Asia. With a rich history of medicinal use dating back to antiquity, *Salix* species have been associated with the discovery of salicylic acid and aspirin. They contain a wide range of flavonoids, including flavones, flavonols, flavanones, dihydroflavonols, isoflavones, chalcones, dihydrochalcones, flavan-3-ols, and anthocyanins, and have traditionally been employed to relieve painful conditions of musculoskeletal joints, inflammation and fever. Salicin, a major pharmacologically active metabolite present in *Salix* species, contributes to its therapeutic properties [66].

Bassino E. et al. (2018) conducted a comparative study to investigate the protective effects of white willow (*Salix alba* L.) bark (standardized for its salicin content) and 1,2-decanediol (an alkanediol that acts as a regulator of the skin homeostasis) on LPS-stressed HaCaT cells in the context of perifollicular inflammation associated with *C. acnes* infection. They found that preincubation of HaCaT cells with the willow bark extract and 1,2-decanediol, either alone or in combination, effectively mitigated LPS-induced cell damage. These effects included the regulation of growth factors (IGF, EGF, VEGF), cytokine production (IL-1α, IL-6, IL-8), and expression of the transcription factor FOXO-I. Additionally, the compounds partially restored the impaired wound repair caused by LPS. These findings suggest that both natural compounds exhibit distinct effects on various functions of LPS-stressed keratinocytes, indicating their potential for the prevention of AV without adverse effects. Salicin (Figure 20), the primary constituent of white will bark extract, is metabolized to salicylic acid in vivo and is known for its anti-inflammatory properties. Moreover, other ingredients in the extracts, such as salicylates, polyphenols, and flavonoids, may also contribute to their therapeutic efficacy [34].

### 3.17. Sapindaceae

The *Sapindaceae* family, also known as the soapberry family, boasts over 1000 species belonging to 125 genera, with a widespread distribution in the tropics and warm subtropics. Although most species are native to Asia, there are also representatives in South America, Africa, and Australia. These plants possess therapeutic potential due to the presence of compounds of pharmaceutical interest such as saponins, which are responsible for their biological activities [67].

Wei M.-P. et al. evaluated the freckle-removing and skin-whitening activities (anti-lipase and anti-tyrosinase assays) of *Sapindus mukorossi* Gaertn. extracts, whose saponin fractions were purified by semi-preparative HPLC and tested against *C. acnes* for their antibacterial activity. The saponin fraction F4, purified from the fermentation liquid-based water extract and rich in four oleanane-type triterpenoids (i.e., Mukurozisaponin E1, Rarasaponin II, Mukurozisaponin G, and Rarasaponin V; Figure 21), showed superior antibacterial activity against *C. acnes* compared to the aqueous crude extract with a MIC value of 0.06 mg/mL and 2.0 mg/mL. The anti-AV potential of the four major compounds was predicted by means of a network pharmacology analysis which indicated the protein coding genes PTGS2 (prostaglandin-endoperoxide synthase 2) and F2RL1 (coagulation factor II receptor-like 1) as main targets, with no toxicity to rats. Both saponin fraction F4 and crude extract were applied to facial masks with no significant influence on their physicochemical properties, suggesting their potential as additives for cosmetic applications [35].

### 3.18. Smilacaceae

The *Smilacaceae* family (or the greenbriers) includes herbaceous to woody vines (e.g., lianas and shrubs) and is widely distributed mainly in tropical and subtropical regions, extending into the temperate zones of both the southern and northern hemispheres. The *Smilax* genus is the only one in the family, and the use of the roots of *S. china* L. is still popular in traditional Korean medicine [68].

Joo J.-H. et al. [36] published a study on the chemical composition and anti-AV properties of extracts derived from the roots of *S. china* L. This study highlighted that among the primary components found in the ethyl acetate-soluble fraction of the plant extracts, resveratrol exhibited superior efficacy in inhibiting the growth of two *C. acnes* strains (KCTC 3314 and KCTC 3320), followed by oxyresveratrol and quercetin (Figure 22), with MIC values of 31.25, 125, and 250 µg/mL, respectively. Besides, these phenolic derivatives are among the most studied compounds as active ingredients in cosmetics for the treatment of skin pathologies [69,70].

### 3.19. Zingiberaceae

The family *Zingiberaceae*, known as the ginger family, encompasses flowering plants and is the largest family in the order Zingiberales. It includes about 56 genera and about 1300 species. These aromatic herbs thrive in the moist regions of the tropics and subtropics, including some areas with seasonal dryness. Renowned for their exceptional biodiversity, the *Zingiberaceae* include some of the most significant and economically valuable plants, characterized by a wide range of colors, shapes and sizes. Furthermore, they have been used for various purposes and their use has been passed down from generation to generation in human cultures, particularly in southeast Asia [71]. Prominent members of this family include ginger (*Zingiber officinale* Roscoe), turmeric (*Curcuma longa* L.), Javanese ginger (*Curcuma zanthorrhiza* Roxb.), and Thai ginger (*Alpinia galanga* L.). The main chemical classes of bioactive compounds that can be found in these plants are gingerols, curcuminoids, and flavonoids, which are known for their remarkable antioxidant, anti-inflammatory, antidiabetic, hepatoprotective, neuroprotective, antimicrobial, and anticancer properties [72].

Sitthichai P. et al. (2022) assessed the antimicrobial and anti-inflammatory activity profile of six different rhizome extracts of *Kaempferia parviflora* Wall., commonly known as black ginger. Among the extracts, nine flavones were detected as the main bioactive compounds, with 5,7-dimethoxyflavone as the predominant component in all but the extract in *n*-hexane, which contained 3,5,7-4′-tetramethoxyflavone as the main component (Figure 23). All extracts were effective against *C. acnes* with MICs in the range of 15–30 µg/mL, whereas only the ethyl acetate extract displayed antimicrobial activity against *S. epidermidis*, probably due to its higher content of total flavonoids. The latter extract was selected for further biological assessments, showing efficacy in inhibiting NO production (IC_50_ = 12.59 µg/mL) with no significant toxicity on fibroblasts. Furthermore, this extract was tested on volunteers in a 0.02% gel-cream formulation, showing notable decrease in the acne severity index (36–52%) and skin erythema (~18%) [37].

## 4. Miscellaneous

Kılıç S. et al. reported the anti-AV efficacy of two mixed plant extracts by assessing their antimicrobial activity against two strains of *C. acnes* (i.e., the reference strain ATCC 51277 and the clinical isolate from a patient), cytotoxicity against human keratinocytes, and performing gene expression analyses with RT-qPCR. Anti-AV extract 1 (AE1) consisted of *Juglans regia* L. (walnut husk), *Myrtus communis* L. (myrtle leaves), *Matricaria chamomilla* L. (chamomilla flowers), *Urtica dioica* L. (stinging nettle leaves), and *Rosa damascena* Herrm. (rose flowers). Anti-acne extract 2 (AE2) contained *Brassica oleracea* var. botrytis L. (broccoli) and *Brassica oleracea* var. italica L. (cauliflower). Both mixed extracts showed outstanding antimicrobial (MICs < 1/2048 µg/mL) and anti-inflammatory activity realistically due to the synergistic effects of the (poly)phenolic constituents with coumarins, tannins, polyacetylenes, and alkaloids. AE1 increased the expression level of TNF-α and suppressed the expression level of IL-1α and SRD5A1 (3-oxo-5α-steroid 4-dehydrogenase 1), whereas AE2 suppressed gene expression level of all three of those [38].

Cohen G. et al. (2023) investigated the anti-AV potential of different plant extracts and plant extract combinations with the addition of cannabidiol, a phytocannabinoid from *Cannabis sativa* that exhibits a promising spectrum of therapeutic actions for the treatment of acne [73]. The objective of this study was that of exploring a possible synergistic effect between the plant extracts and the added active component by targeting different pathogenic factors of acne and minimizing side effects at the same time. The initial phase of the study entailed the capacity of these combinations to inhibit *C. acnes* growth and reduce IL-1β and TNF-α secretion from U937 cells. Results revealed that a combination of *Centella asiatica* triterpene extract, rich in madecassoside, asiaticoside, madecassic acid, and asiatic acid (Figure 24), and silymarin (a standardized extract from *Silybum marianum* fruits) exhibited significantly higher antimicrobial and anti-inflammatory activity when combined with cannabidiol than each component alone. Moreover, in an ex vivo experimental model (human skin organ cultures), these three ingredients combined in a topical formulation showed efficacy in reducing pro-inflammatory cytokine IL-6 and IL-8 hypersecretion without hampering epidermal viability. The formulation turned out to be effective in a preliminary clinical study, significantly reducing acne lesions and porphyrin levels [39].

Another study on miscellaneous natural products was carried out by Mias C. et al. [40]. The authors evaluated the single and combined pharmacological properties of Myrtacine^®^, an extract from *Myrtus communis* L. (Celastraceae), and celastrol (a triterpenoid secondary metabolite isolated from the roots of *Tripterygium wilfordii* Hook. f. and *Tripterygium regelii* Sprague & Takeda) enriched plant cell culture extracts (CEEs) against *C. acnes* phenotype IA1, and investigated the pathways linked to acne on Th17 lymphocytes by means of a 2D model of *C. acnes*-stimulated sebocytes integrated in a 3D skin model. The obtained results indicated that both Myrtacine^®^ and CEEs significantly inhibit the production of pro-inflammatory cytokines such as IL-6, IL-8, IL-12p40, and TNF-α in *C. acnes*-stimulated monocytes-derived dendritic cells and that the effect is enhanced when they act in combination. The 2D/3D experiments indicated that CEEs, in solution or 0.3% formulation, inhibit IL-17 release by Th17 cells, suggesting that one or more of these pathways are affected.

In the study of Oliveira A.S. et al. (2023), two Portuguese autochthonous plant species traditionally used in folk medicine for skin applications were investigated, i.e., *Thymus mastichina* (L.) L. and *Cistus ladanifer* L. The main components identified in the EOs of *T. mastichina* and *C. ladanifer* were 1,8-cineole (see Figure 3) and α-pinene (see Figure 5), respectively. In the corresponding hydrolates instead, 1,8-cineole and (*E*)-pinocarveol were the predominant compounds. The *C. ladanifer* EO exhibited the strongest anti-inflammatory potential (NO production assay) and broad-spectrum antimicrobial (including *C. acnes*) activity, although associated with significant cytotoxicity on RAW 264.7 and L929 cell lines. *T. mastichina* preparations also displayed significant anti-inflammatory properties with better biocompatibility. Both the EO and hydrolate of *C. ladanifer* increased fibroblasts’ migration, while the *T. mastichina* hydrolate, although less potent than the *C. ladanifer* hydrolate, still promoted wound healing by increasing cell migration. Limited antioxidant capacity was detected for all extracts [41].

Sonyot W. et al. (2020) reported the antibacterial and anti-inflammatory effects of components derived from the entomopathogenic fungus *Polycephalomyces phaothaiensis*. This fungus, upon cultivation in potato dextrose agar broth, was extracted to provide four crudes from which were isolated two known tropolone derivatives, namely cordytropolone and stipitalide, and five known compounds, namely (+)-piliformic acid, D-mannitol, methyl linoleate, linoleic acid, and ergosterol. The anti-*C. acnes* activity of the extracts and individual compounds was evaluated using both agar diffusion and broth dilution assays. Results revealed that the ethyl acetate extract and the two tropolone derivatives displayed considerable antibacterial activity against *C. acnes* (MICs 8–64 µg/mL). In contrast, (+)-piliformic acid exhibited weaker inhibitory effects. Subsequently, the anti-inflammatory properties of the ethyl acetate extract and compounds cordytropolone, stipitalide, and (+)-piliformic acid (Figure 25) were assessed by quantifying the pro-inflammatory cytokines IL-1β, IL-6, and TNF-α in THP-1 cells stimulated with heat-killed *C. acnes*. The findings demonstrated a significant and potent inhibitory effect of the extract and its constituents on the production of *C. acnes*-induced pro-inflammatory cytokines in THP-1 cells. These results suggest, for the first time, the therapeutic potential of *P. phaothaiensis* and its constituents cordytropolone and stipitalide as adjuvants in the treatment of AV [42].

## 5. Discussion

Plants and their extracts have a long history in medical folk tradition by virtue of the many bioactive components they contain. In relation to the anti-AV properties of phytochemicals, numerous studies have reported that their antimicrobial activity is associated with different mechanisms of action, such as induction of ROS production, inhibition of cell wall synthesis which eventually causes cell lysis, inhibition of biofilm formation, block of DNA replication, inhibition of energy production, and inhibition of synthesis of bacterial toxins to the host (Figure 26). In addition, these compounds can prevent antibiotic resistance and/or have synergistic effects with antibiotics [74].

From what has been presented in this review, it is evident that (poly)phenolic compounds play a primary role in performing all of these biological activities. Their properties of scavenging ROS; their antimicrobial, antiproliferative, and photoprotective properties; and their properties of activating antioxidant enzymes, chelating metals, inhibiting oxidases, attenuating oxidative stress caused by NO, and enhancing the antioxidant properties of low molecular weight antioxidants have been extensively documented in the recent scientific literature. In particular, (poly)phenolic compounds (especially flavonoids) have also shown anti-inflammatory properties and have been shown to ameliorate acute or chronic inflammation by reducing oxidative stress and pro-inflammatory states [3]. Their anti-inflammatory effects can be exerted through several mechanisms: they can affect cellular signaling pathways, including NF-κB, MAPK, and PI3K/Akt, or inhibit several key regulators of the inflammatory immune response, such as TNF-α, IL-1β, and IL-6. In addition, their anti-inflammatory activity can be carried out at the molecular level through the inactivation of pro-inflammatory enzymes such as LOX and COX [75]. The issues of pharmacokinetics (e.g., rapid first pass metabolism) and the poor bioavailability of this type of compounds are also well documented, a fact that has led to extensive studies into the development of effective topical delivery systems [76,77,78]. Similarly, compounds in EOs (which should never be used pure as they can trigger strong allergic reactions and cause skin irritation) have shown intriguing activity against AV. Indeed, several studies have attempted to elucidate the mechanisms of action of EOs; however, these mechanisms still remain unclear. Mainly, it has been observed that some constituents of EOs penetrate the peptidoglycan layer, disrupting the cytoplasmic membrane of bacteria, and causing the cytoplasmatic contents to leak out. Their action thus results mainly in bactericidal and anti-biofilm action and is related to the high lipophilicity of the bioactive compounds [10,79]. A complete list of proven mechanisms of action of the phytochemicals reported in this review article is presented in Table 2 (see hereinafter).

Because acne is a skin condition with a multifactorial etiology, its therapeutic treatment is quite complex and often requires combination therapies that are still evaluated according to the severity of the disease or skin lesions [90]. Contrary to common thinking, treatment of AV with antibiotic drugs is not the drug therapy of first choice because, in many cases, bacterial infection is a consequence of acne and not the primary cause. In fact, as we have said, the appearance of acne can be traced to the obstruction of the hair follicle caused by sebum (often produced in excess) and cellular debris. The obstruction can then give rise to inflammatory processes that are exacerbated by bacterial infections, usually sustained by microorganisms that under normal conditions live on the skin without causing problems. However, when the right conditions occur, such microorganisms can proliferate uncontrollably, giving rise to infections. The current pharmacological treatments of acne have advantages and disadvantages. For example, oral isotretinoin (i.e., 13-*cis*-retinoic acid), the use of which is recommended in cases of severe nodulocystic acne, is teratogenic; some antibiotics (e.g., clindamycin and erythromycin) are effective, but with the spread of bacterial resistance there is a risk of progressive loss of efficacy, so they are often used in combination therapy to reduce their dosage and decrease the possibility of the emergence of bacterial resistance. In this context, the use of medicinal plants has considerable potential to address the multifaceted challenges associated with acne treatment and is an area that is constantly evolving in terms of efficacy and applicability. Against acne, there are already good phytotherapeutic alternatives, distinguishable by type of action into systemic remedies with draining, purifying and anti-inflammatory action and topical remedies with moisturizing and disinfectant action for the skin. These include burdock (*Arctium lappa*), whitethorn (*Crataegus monogyna*), and wild pansy (*Viola tricolor*) extracts, and tea tree oil. This review has shown that several compounds derived from other plants are quite effective for the treatment of mild AV or can be used in combination with conventional therapy as adjuvants. The main goal of research is now to optimize their therapeutic profile through appropriate formulations. Overall plant-based medicines/remedies offer a compelling alternative to the standard treatment of acne with minimal or negligible side effects. However, although many of them have demonstrated efficacy and safety profiles through clinical trials, they cannot yet be called replacement therapies, so scientific research interest in this area remains keen [91].

## Figures and Tables

**Figure 1 molecules-29-02394-f001:**
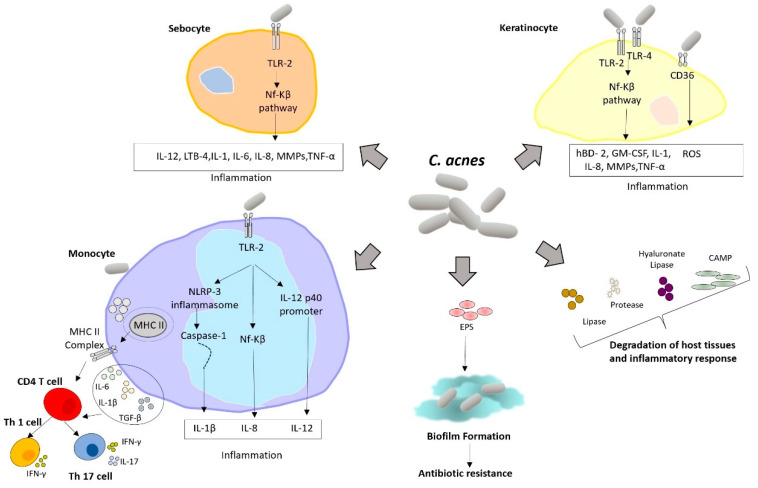
A proposed model of the primary pathological processes induced by *C. acnes* involves the interplay between sebocytes, keratinocytes, and monocytes in acne vulgaris. MHC II: major histocompatibility complex II; Th17: cell T helper 17 cell; IL: interleukin; TGF: transforming growth factor; TLR: toll-like receptor; IFN-γ: interferon-gamma; CAMP: cyclic adenosine monophosphate; CD36: cluster of differentiation 36; EPS: extracellular polymeric substances; GM-CSF: granulocyte-macrophage colony stimulating factor; hBD: human β-defensin; LTB: leukotriene B; MMPs: matrix metalloproteinases; TLR: Toll-like receptor; TNF: tumor necrosis factor.

**Figure 2 molecules-29-02394-f002:**
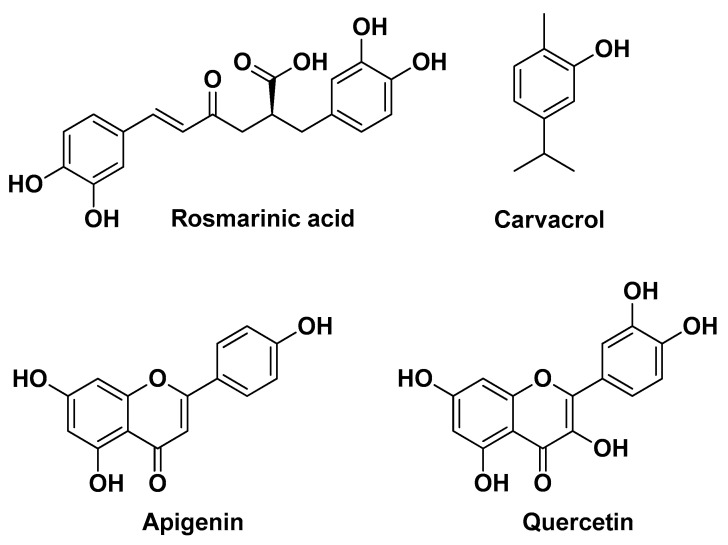
The four major bioactive compounds identified by Chuang L.-T. et al. from *Origanum vulgare* L. leaf extracts [8].

**Figure 3 molecules-29-02394-f003:**
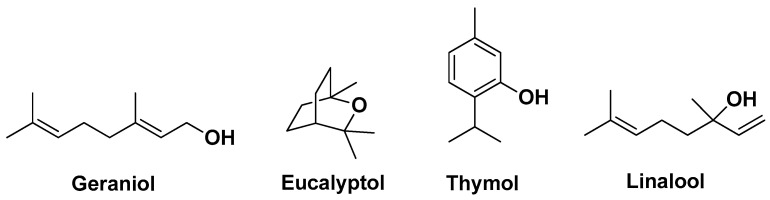
The four major bioactive monoterpenoids identified by Oliveira A.S. et al. from EO and hydrolate of *Thymus* × *citriodorus* (Pers.) Schreb [11].

**Figure 4 molecules-29-02394-f004:**
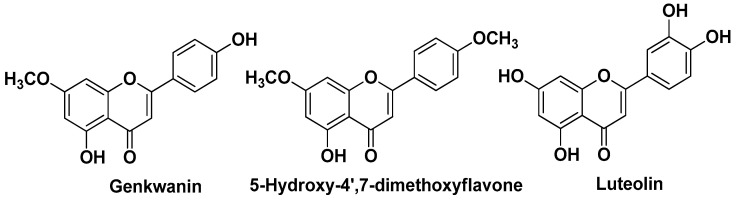
The main flavonoids isolated by Pineau R.M. et al. from ethanol leaf extracts of *Callicarpa americana* L. [12].

**Figure 5 molecules-29-02394-f005:**
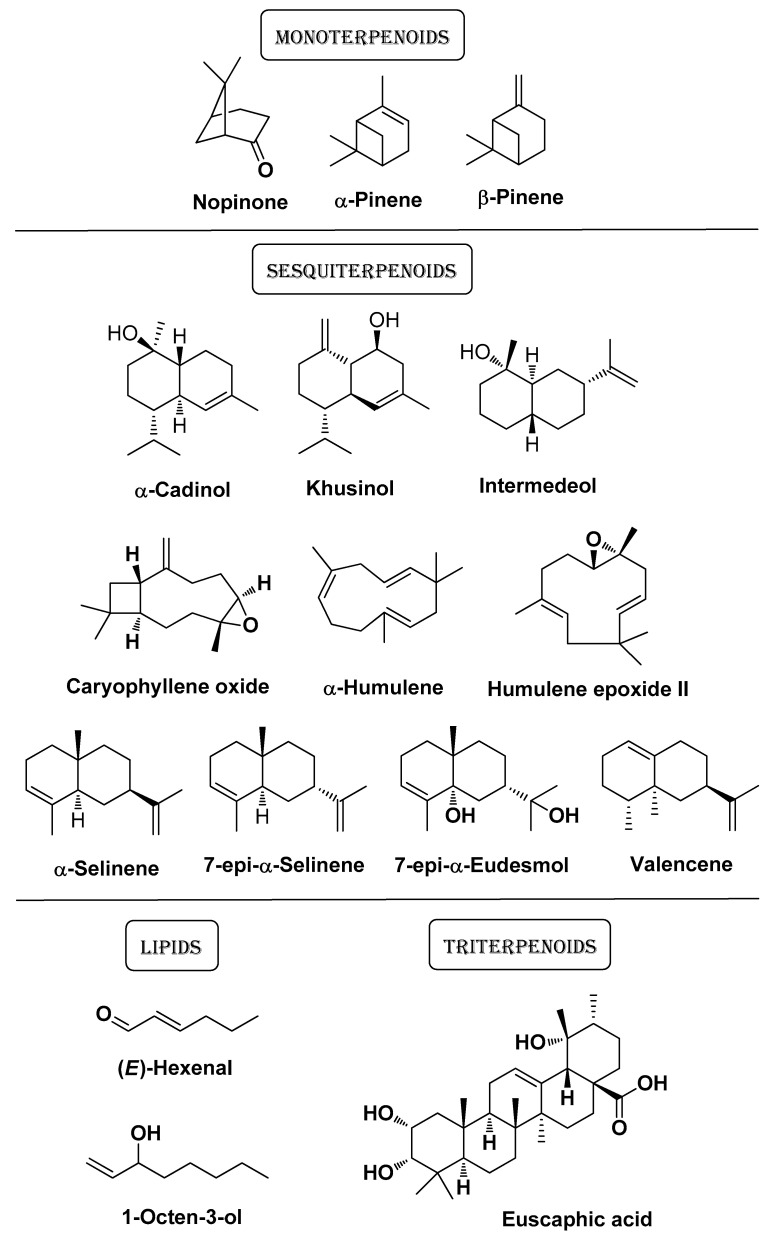
The main bioactive compounds isolated by Pineau R.M. et al. from leaf EOs of *Callicarpa americana* L. [12].

**Figure 6 molecules-29-02394-f006:**
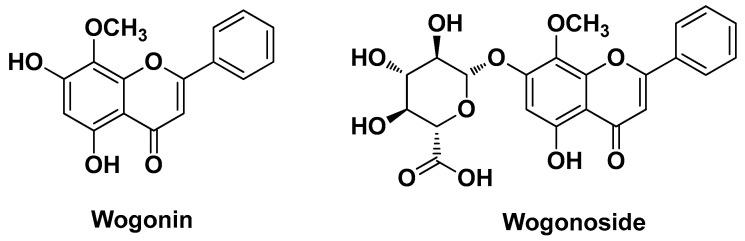
Chemical structure of the flavonoid wogonin and its glycoside (wogonoside) identified by Zhu X. et al. from *Scutellaria baicalensis* Georgi [14].

**Figure 7 molecules-29-02394-f007:**
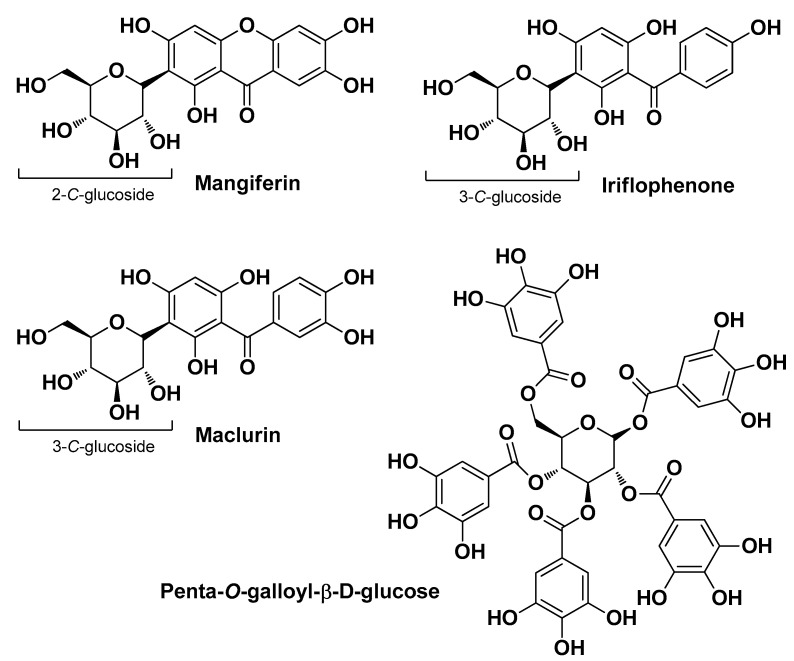
The four main bioactive compounds isolated by De Tollenaere M. et al. from *Mangifera indica* L. leaf extracts [16].

**Figure 8 molecules-29-02394-f008:**
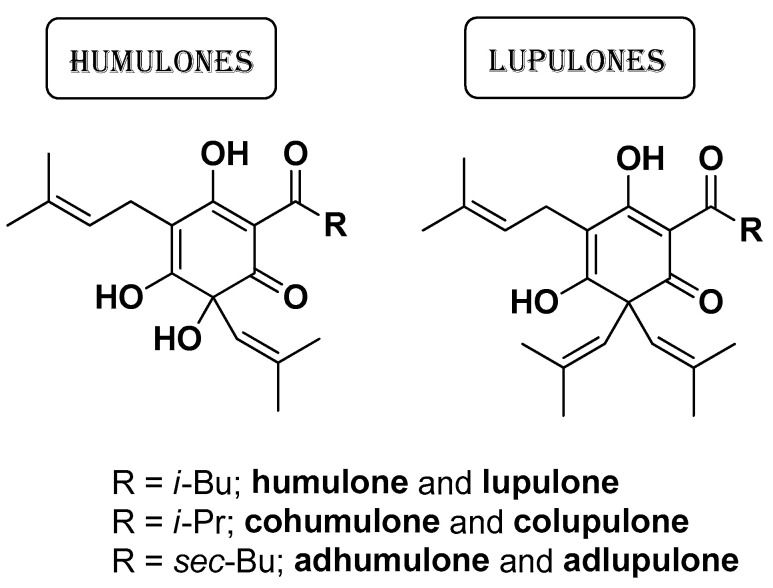
Main bioactive compounds detected by Weber N. et al. in a hop-CO_2_ extract derived from flowers of *Humulus lupulus* [19].

**Figure 9 molecules-29-02394-f009:**
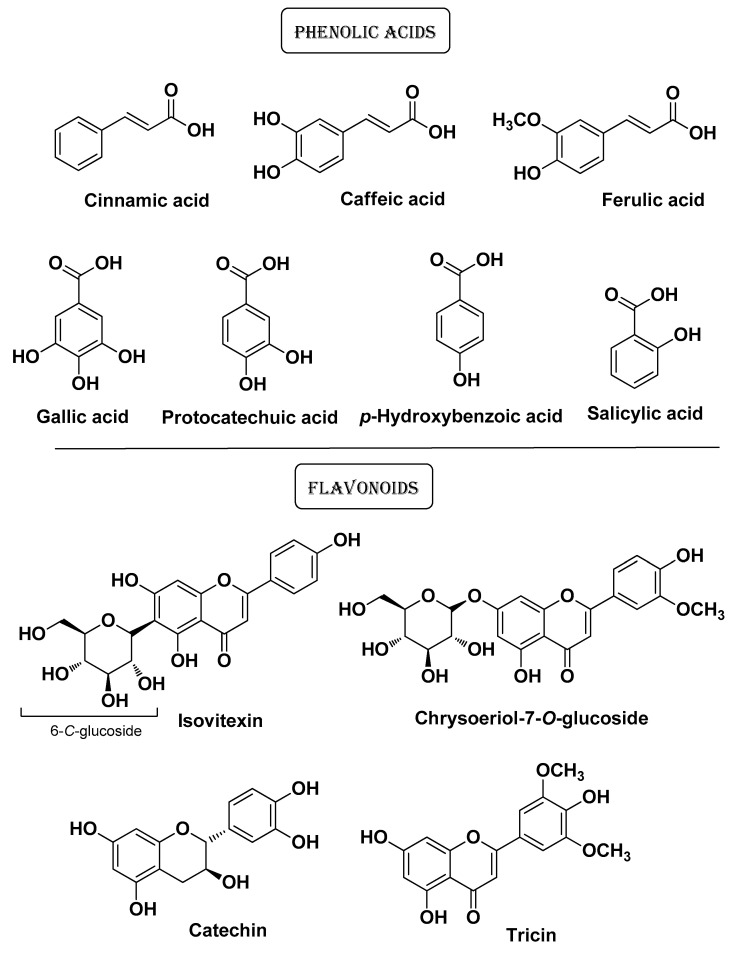
Some of the main bioactive compounds isolated by Kim C. et al. from *Cymbopogon citratus* Stapf. Extracts [20].

**Figure 10 molecules-29-02394-f010:**
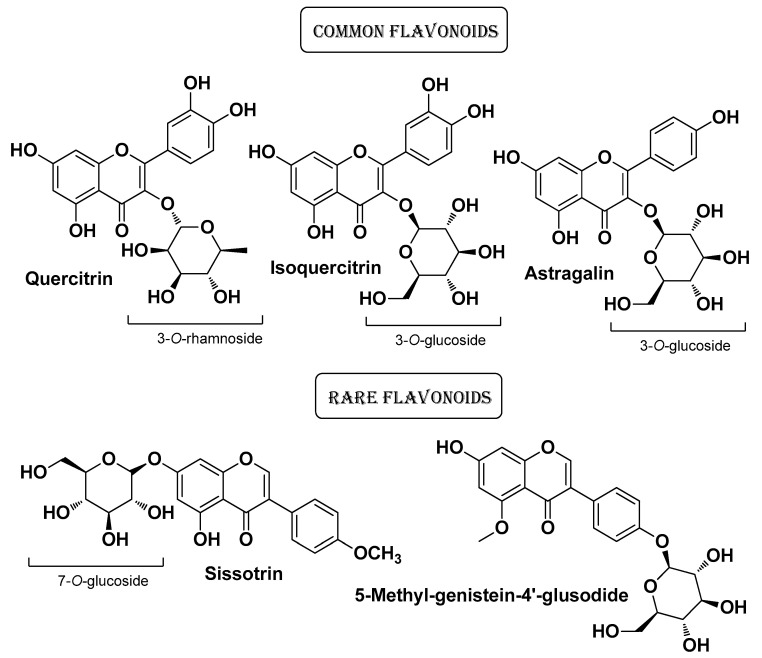
The most abundant flavonoids identified by Krzemińska B. et al. from *Cotoneaster nebrodensis* and *Cotoneaster roseus* Collett extracts [23].

**Figure 11 molecules-29-02394-f011:**
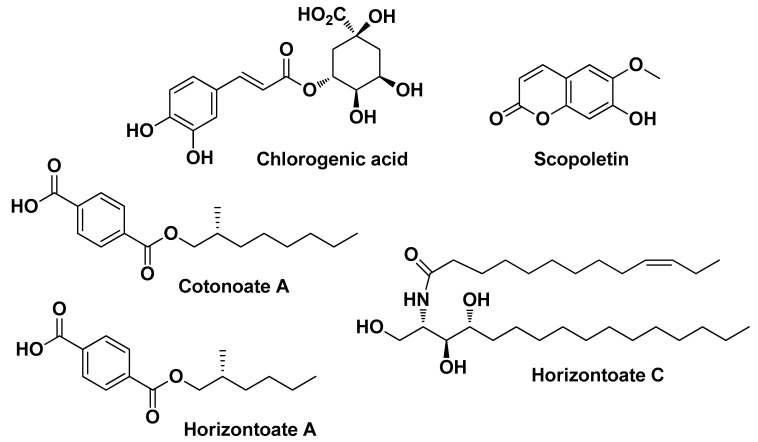
The most abundant bioactive compounds (flavonoids aside) identified by Krzemińska B. et al. from *Cotoneaster nebrodensis* (Guss.) K. Koch and *Cotoneaster roseus* Collett extracts [23].

**Figure 12 molecules-29-02394-f012:**
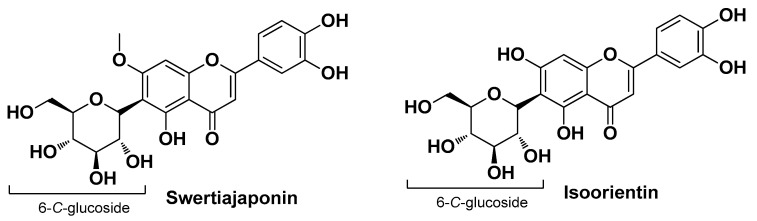
Two of the most abundant bioactive flavonoids identified by Chrząszcz M. et al. from *Cephalaria uralensis* (Murray)Roem. & Schult and *Cephalaria gigantea* (Ledeb.) Bobrov extracts [25].

**Figure 13 molecules-29-02394-f013:**
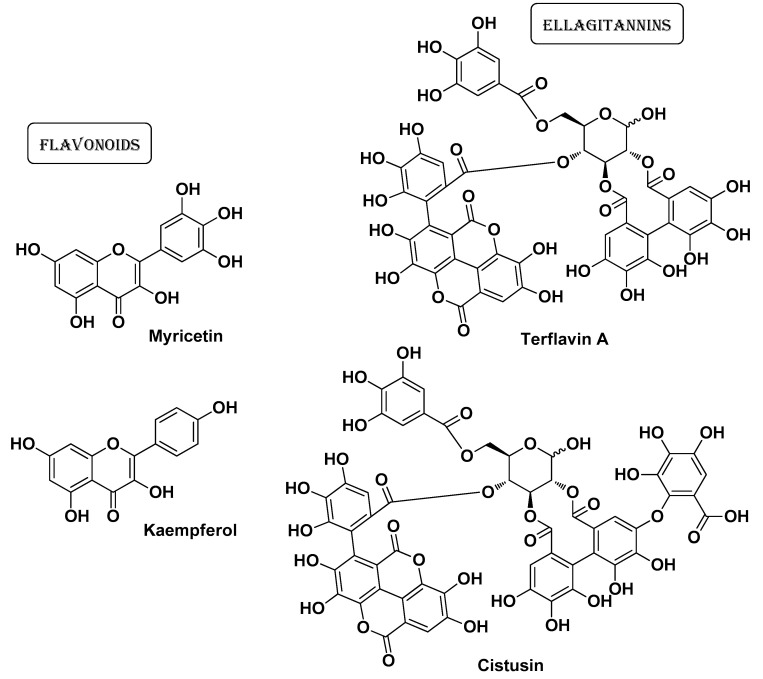
Four of the most abundant bioactive compounds identified by Bouabidi M. et al. from *Cistus laurifolius* L. and *Cistus salviifolius* L. [26].

**Figure 14 molecules-29-02394-f014:**
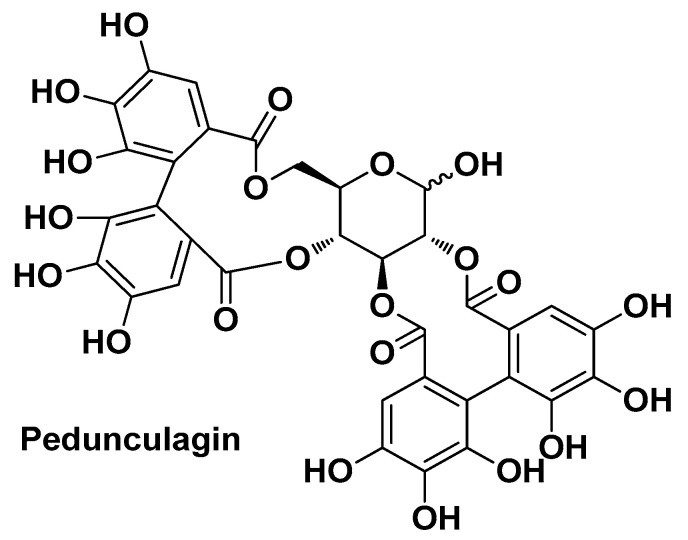
Chemical structure of pedunculagin, the main bioactive compound identified by Kim M. et al. from *Quercus mongolica* Fisch [27].

**Figure 15 molecules-29-02394-f015:**
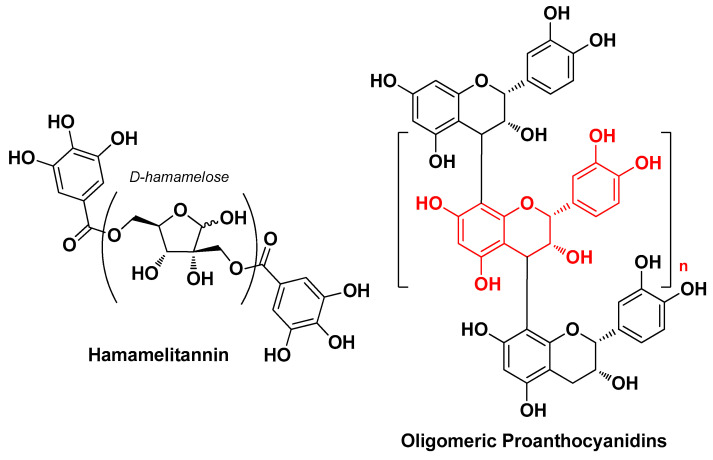
Most abundant phytochemicals identified by Piazza S. et al. in the glycolic extract of *Hamamelis virginiana* L. bark (n = 0–7) [28].

**Figure 16 molecules-29-02394-f016:**
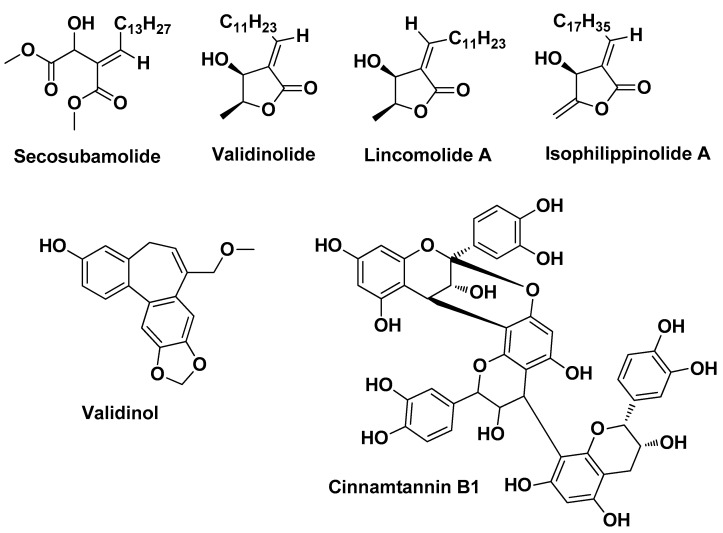
The six most active derivatives isolated from the stem of *Cinnamomum validinerve* and characterized by Yang C.L. et al. [29].

**Figure 17 molecules-29-02394-f017:**
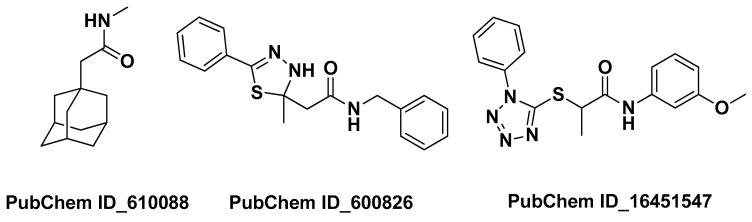
The three bioactive components identified by Kola-Mustapha A.T. et al. in the extract of *Azadirachta indica* A. Juss. that may target genes in the treatment of AV [30].

**Figure 18 molecules-29-02394-f018:**
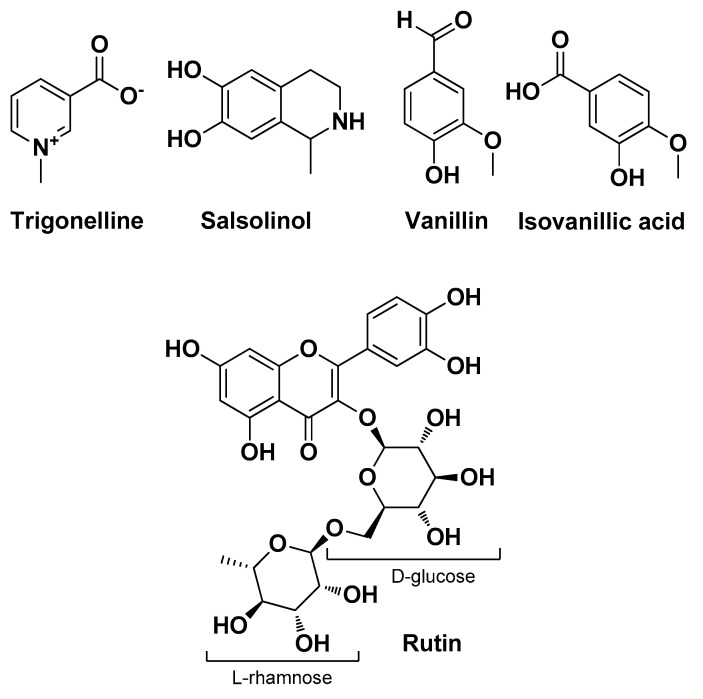
Some of the main bioactive compounds isolated by Savitri D. et al. from *M. balbisiana* Colla peel extracts [31].

**Figure 19 molecules-29-02394-f019:**
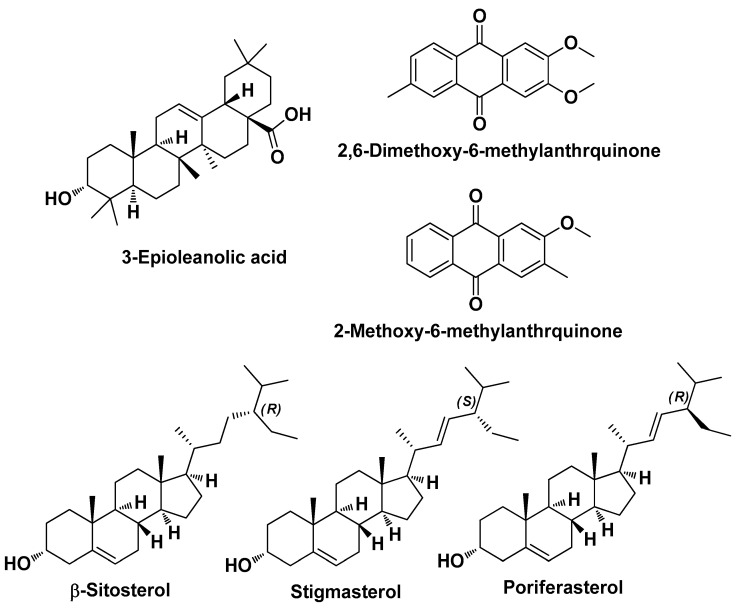
Six out of the seven hit compounds identified by Seo G. and Kim K. in *Hedyotis diffusa* Willd (the other being quercetin, already depicted in Figure 2) using network analysis [33].

**Figure 20 molecules-29-02394-f020:**
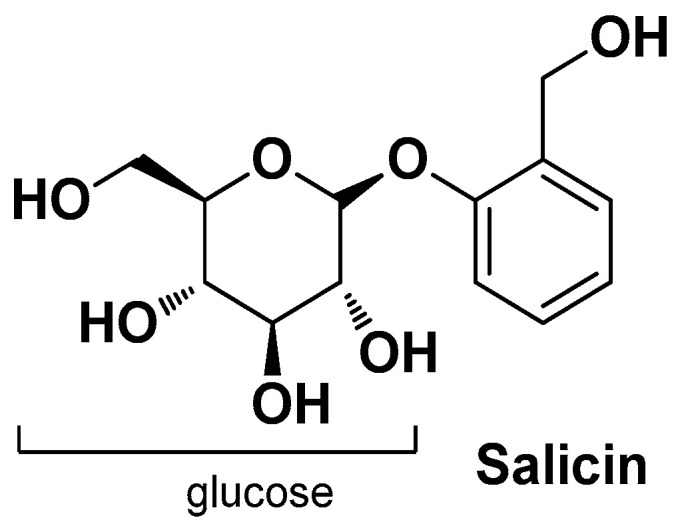
Chemical structure of salicin, the major bioactive component white willow (*Salix alba* L.) bark [34].

**Figure 21 molecules-29-02394-f021:**
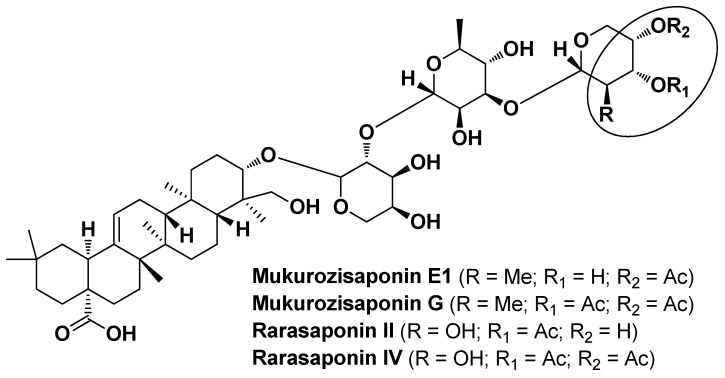
Chemical structure of most of the saponins identified by Wei et al. in *Sapindus mukorossi* Gaertn. extracts [35].

**Figure 22 molecules-29-02394-f022:**
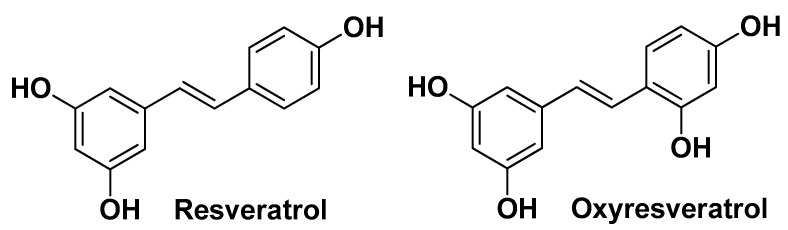
The chemical structure of two of the most active compounds identified by Joo J.-H. et al. in *Smilax china* L. root extracts (the third most abundant being quercetin, already depicted in Figure 2) [36].

**Figure 23 molecules-29-02394-f023:**
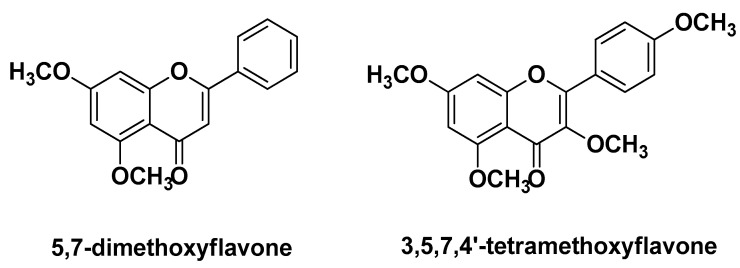
Chemical structure of two of main compounds identified by Sitthichai P. et al. in *Kaempferia parviflora* Wall. rhizome extracts [37].

**Figure 24 molecules-29-02394-f024:**
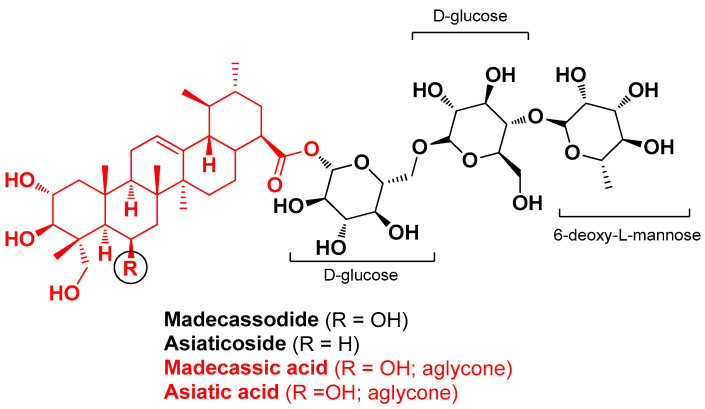
Main triterpenoids identified by Cohen G. et al. in *Centella asiatica* triterpene extract [39].

**Figure 25 molecules-29-02394-f025:**
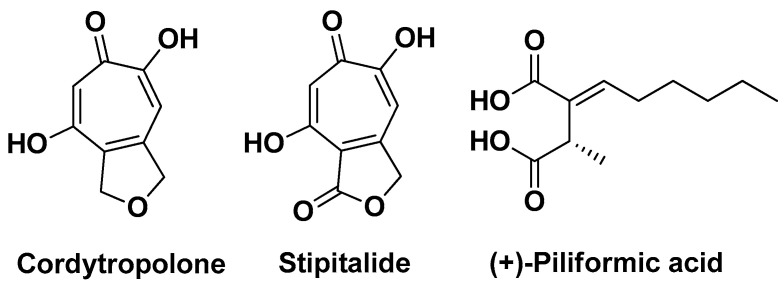
The chemical structure of the most bioactive compounds identified by Sonyot W. et al. in the culture broth extract of the fungus *Polycephalomyces phaothaiensis* [42].

**Figure 26 molecules-29-02394-f026:**
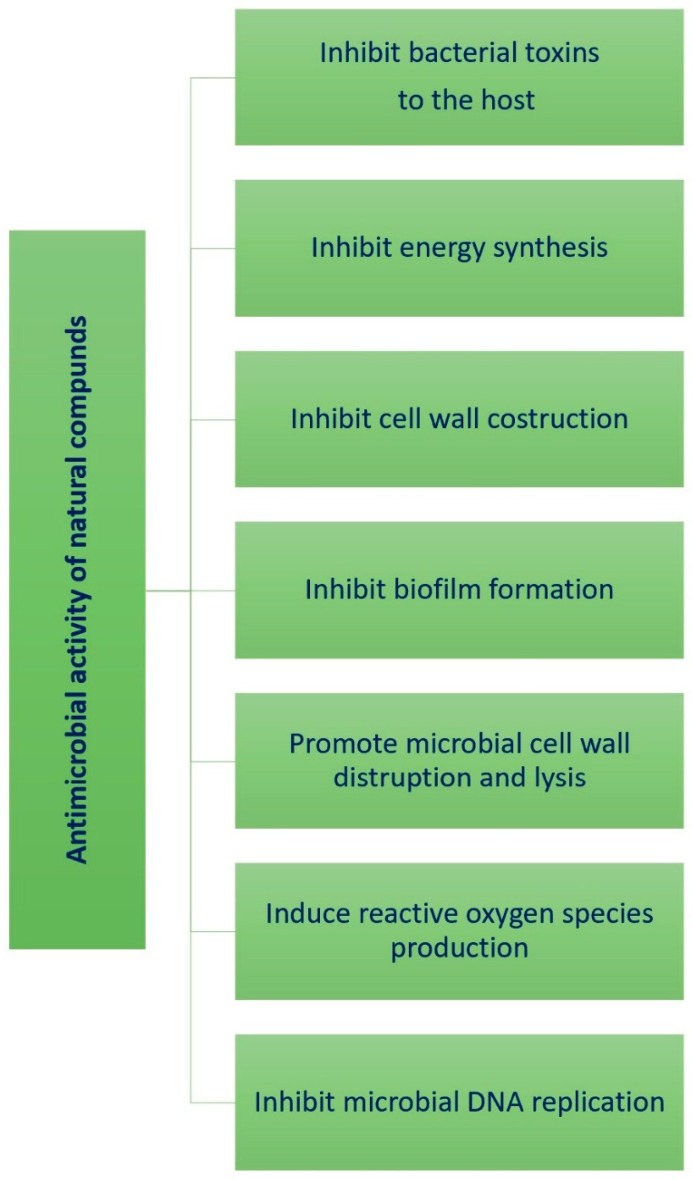
Mechanisms of antimicrobial activity of compounds from medicinal plants.

**Table 1 molecules-29-02394-t001:** Main characteristics of the studies included in this review.

Natural Source	Common Name	Part Used	Active Compounds	Reported Biological Activity	Ref.
*Origanum vulgare* L. ***Lamiaceae***	Oregano	Leaves (extracts)	Rosmarinic acid Quercetin Luteolin Apigenin Carvacrol Thymol	Anti-inflammatory	[8]
*Origanum vulgare* L. ***Lamiaceae***	Oregano	Essential oil	Thymol	Antimicrobial Anti-inflammatory	[9]
*Thymus vulgaris* L. ***Lamiaceae***	Thyme	Essential oil (Nanoemulsion)	Thymol Caryophyllene Phenolic compounds Terpenoid compounds	Antimicrobial Anti-inflammatory	[10]
*Thymus* × *citriodorus* *(Pers.) Schreb.* *(hybrid Thymus* *pulegioides* L. and *Thymus vulgaris* L. ***Lamiaceae***	Lemon thyme	Essential oil Hydrolate	Geraniol 1,8-Cineole Thymol Linalool	Antimicrobial Anti-biofilm Anti-inflammatory	[11]
*Callicarpa americana* L. ***Lamiaceae***	American beautyberry	Leaves (extracts)	Six clerodane diterpenes Genkwanin 5-Hydroxy-7,4′-dimethoxyflavone Luteolin	Antimicrobial Anti-inflammatory	[12]
*Plectranthus aliciae* (Codd) van Jaarsv. & T.J. Edwards ***Lamiaceae***		AuNP of Leaves and soft twigs (extracts)	Rosmarinic acid	Antimicrobial and Wound healing potential	[13]
*Scutellaria baicalensis* Georgi ***Lamiaceae***	Chinese skullcap	Aereal part (extract)	Baicalin Wogonoside, Lincomolide A Secosubamolide Cinnamtannin B1 Isophilippinolide A Secosubamolide	Anti-inflammatory Antimicrobial	[14]
*Mangifera indica* L. ***Anacardiaceae***	Mango	Raw and ripe fruits (extracts)	Gallic acid	Antioxidant Anti-inflammatory	[15]
*Mangifera indica* L. ***Anacardiaceae***	Mango	Leaf (extract)	Mangiferin (glucosylxanthone -xanthonoid) Penta-*O*-galloyl-beta-D-glucose Iriflophenone-3-C-beta-glucoside Maclurin-3-C-beta-glucoside	Sebo regulation Antimicrobial	[16]
*Anacardium occidentale* L. ***Anacardiaceae***	Cashew	Peduncle pulp (extract)	Rutin	Antioxidant Antimicrobial	[17]
*Cannabis sativa* L. ***Cannabaceae***	Hemp	Seed (extracts)	Linoleic acid Oleic acid *cis*-11-Eicosenoic acid Palmitic acid γ-Linolenic acid Arachidic acid, Palmitoleic acid Heneicosanoic acid	Anti-inflammation Anti-lipogenesis	[18]
*Humulus lupulus* L. ***Cannabaceae***	Hop	Hop-CO_2_-extract	Humulones Lupulones	Antioxidant Anti-inflammatory Antimicrobial	[19]
*Cymbopogon citratus* Stapf ***Poaceae***	Lemongrass	Aereal part (extracts)	Caffeic acid Salicylic acid *p*-Hydroxybenzoic acid, Gallic acid Ferulic acid Isovitexin Luteolin Catechin Tricin Protocatechuic acid, Chrysoriol 7-*O*-glucoside Catechin k	Antioxidative, Antimicrobial Anti-aging Anti-whitening	[20]
*Zea mays* L. ***Poaceae***	Corn	Biosurfactant extract obtained from corn milling industry (named BS-CSW)	Salycilic acid	Antimicrobial	[21]
*Cotoneaster hsingshangensis J. fryer & B. hylmö* and *Cotoneaster issaricus Pojatk* ***Rosaceae***		Leaves (extracts)	Isoquercitrin Rutin hyperoside Quercitrin Chlorogenic acid Gentisic acid 2-*O*-glucoside Scopoletin	Antioxidant Anti-cyclooxygenase Anti-lipoxygenase Anti-hyaluronidase Antimicrobial	[22]
*Cotoneaster nebrodensis* (Guss.) K. Koch and *Cotoneaster roseus* Coll ***Rosaceae***	Brickberry cotoneaster, Madagascar periwinkle	Leaves and fruits (extract)	Flavonoids (quercetin derivatives)	Anti-lipoxygenase, Anti-hyaluronidase, Anti-cyclooxygenase Antimicrobial	[23]
*Arctium lappa* L. ***Asteraceae***	Burdock	Roots (extract)	Peptides (Br-p) isolated	Antimicrobial Antioxidant	[24]
*Cephalaria uralensis* Roem. & Schult. and *Cephalaria gigantea* (Ledeb.) Bobrov ***Caprifoliaceae***	Murray and giant scabious	Aerial parts and flowers of Murray and the aerial parts of Giant scabious (extract)	5-*O*-Caffeoylquinic acid Isoorinetin Swertiajaponin	Antioxidant Anti-inflammatory Antimicrobial	[25]
*Cistus* *laurifolius* L. and *Cistus salviifolius* L. ***Cistaceae***		Aereal part (extracts)	Myricetin Quercetin Kaempferol Terflavin A Cistusin	Antioxidant Anti-Inflammatory Antimicrobial	[26]
*Quercus mongolica* Fisch. ***Fagaceae***	Mongolian oak	Leaves (extract)	Pedunculagin	Anti-inflammatory 5α-Reductase inhibition	[27]
*Hamamelis virginiana* L. ***Hamamelidaceae***	American witch hazel	Bark (extract)	Hamamelitannin Gallotannins Flavonols Proanthocyanidins	Antioxidant Anti-inflammatory Antimicrobial	[28]
*Cinnamomum validinerve* Hance ***Lauraceae***	Cinnamomum	Stem (extract)	Validinol Validinolide Butanolide Tannins	Anti-inflammatory Antimicrobial	[29]
*Azadirachta indica* A.Juss. ***Meliaceae***	Neem	Leaves (oil)	2-(1-Adamantyl)-*N*-methylacetamide *N*-benzyl-2-(2-methyl-5-phenyl-3*H*-1,3,4-thiadiazol-2-yl)acetamide) *N*-(3-methoxyphenyl)-2-(1-phenyltetrazol-5-yl)sulfanylpropanamide PubChem ID_610088, PubChem ID_600826 PubChem ID_16451547	Anti-inflammatory	[30]
*Musa balbisiana* Colla ***Musaceae***	Weet wild banana	Banana peels (extract)	Rutin	Anti-inflammatory Antimicrobial	[31]
*Meconopsis quintuplinervia* Regel ***Papaveraceae***		(extract)	Alkaloids Flavonoids (quercetin and luteolin) Volatile oils	Antimicrobial	[32]
*Hedyotis diffusa* Willd ***Rubiaceae***	snake-needle grass	(extract)	2-Methoxy-3-methyl-9,10-anthraquinone 2 3-Dimethoxy-6-methyanthraquinone Quercetin Beta-sitosterol Poriferasterol Stigmasterol 3-epioleanolic acid	Sebo reducent Anti-inflammatory	[33]
*Salix alba* L. ***Salicaceae***	White willow	Bark (extract)	Salicilin 1,2-Decanediol (beta glucoside)	Anti-inflammatory	[34]
*Sapindus mukorossi* Gaertn. ***Sapindaceae***	Chinese soapberry	Peel (extract)	Saponin fraction (F4): Mukurozisaponin E1 Rarasaponin II Mukurozisaponin G Rarasaponin VI	Antimicrobial	[35]
*Smilax china* L. ***Smilacaceae***	China root	Root (extract)	Quinic acid Caffeic acid Polydatin Quercetin Oxyresveratrol Catechin Resveratrol	Antimicrobial	[36]
*Kaempferia parviflora* Wall. ***Zingiberaceae***	Thai ginseng	Rhizomes (extracts)	5-hydroxy-7-methoxyflavone, 5-hydroxy-3,7-methoxyflavone 5,7-dimethoxyflavone 5-hydroxy-3,7,40-methoxyflavone	Antimicrobial Anti-inflammatory	[37]
*Juglans regia* L., ***Juglandaceae***; *Myrtus* *Communis* L., ***Myrtaceae***; *Matricaria chamomilla* L., ***Asteraceae***; *Urtica dioica* L., ***Urticaceae***; *Rosa damascena Herrm.*, ***Rosaceae***; *Brassica oleracea var. botrytis* L. ***Brassicaceae***, and *Brassica oleracea var. italica* L. ***Brassicaceae***	Walnut husk myrtle, chamomilla, stinging nettle, rose; broccoli, cauliflower	Anti-acne extract 1 (AE1): walnut husk, myrtle leaves, chamomilla flowers, stinging nettle leaves and rose flowers; Anti-acne extract 2 (AE2): broccoli and cauliflower	Main in AE1: Chlorogenic acid Caffeic acid Ferulic acid Vanillic acid catechin Juglone herbaceous (naftalenedione) Apigenin Rutin Coumarins Polyacetylenes Bisabolol Present in AE2: Alkaloids Carbohydrates Glycosides Tannins Quercetin Kaempferol	Antimicrobial Anti-inflammatory	[38]
*Centella asiatica **Apiaceae*** and *Silybum marianum* L. ***Asteraceae***, *Lonicera japonica flower Caprifoliaceae*, *Salvia miltiorrhiza **Lamiaceae*** and *Camellia sinensis* L. ***Theaceae***; *Salix babylonica* L. ***Salicaceae***	Gotu kola, milk thistle, honey suckle; red sage, green tea, white willow bark	*C. asiatica* triterpene leaf (extract); *S. marianum* fruit (extract); *S. miltiorrhiza* root (extract); *C. sinensis* (extract); *S. babylonica* (extract)	Cannabidiol Asiaticoside Asiatic acid Madecassic acid Silymarin (as silibinin: silicristin, silibinin A and B and isosilibinin A and B); Caffeine	Anti-inflammatory Antimicrobial	[39]
*Myrtus communis* L. ***Myrtaceae*** and *Tripterygium wilfordii* ***Celastraceae***	Myrtle and thunder god vine	*M. communis* L. (extract) (Myrtacin^®^) Celastrol (enriched extract)	Myrtucummulones Ursolic acid Terpenoids Alkaloids Steroids	Anti-inflammatory	[40]
*Thymus mastichina* L. ***Lamiaceae*** and *Cistus ladanifer* L. ***Cistaceae***	White thyme and gum rockrose	Essential oil Hydrolated	CL EO: α-pinene and camphene TM EO: 1,8-cineole, *p*-cymene In both EO: sesquiterpene hydrocarbons and oxygen-containing sesquiterpenes	Antioxidant Anti-inflammatory Wound healing Antimicrobial	[41]
*Polycephalomyces phaothaiensis*	Fungi	extracts	Cordytropolone Stipitalide (+)-piliformic acid	Anti-inflammatory, Antimicrobial	[42]

**Table 2 molecules-29-02394-t002:** Main mechanisms of action of phytochemicals.

Mechanisms		Compounds	Refs.
**Antimicrobial**	Disintegration of bacterial outer membrane or phospholipid bilayer Increase of membrane fluidity with leakage of potassium ions and protons Complexes with cholesterol membrane with increase in permeability and leakage of cytoplasmic contents Inhibition of peptidoglycan synthesis	Apigenin Catechin Ferulic acid Caffeic acid Chrysoriol 7-*O*-glucoside Gallic acid Kaempferol Isovitexin Luteolin Naringenin *p*-Hydroxybenzoic acid Protocatechuic acid Rhamnetin Rosmarinic acid Quercetin Salicylic acid Tricin β-Caryophyllene *p*-Cymene Carvacrol Linalool Menthol Thymol Humulones Lupulones Saponins	[19,20,21,80,81,82,83]
	Inhibition of nucleic acid synthesis or cell envelope synthesis or fatty acid synthase or ATP synthase	Apigenin Baicalein Catechin Kaempferol Luteolin Myricetin Naringenin Quercetin	[80]
	Inhibition of bacterial virulence	Quercetin glycoside Kaempferol	[80]
	Inhibition of efflux pumps	Catechin Genestein Quercetin	[80]
	Alteration of fatty acid composition	Carvacrol Thymol	[84]
	Interference with glucose uptake	Humulones Lupulones	[19,85]
	Interference with oxidative phosphorylation or oxygen uptake	Carvacrol Linalool	[81]
**Anti-inflammatory**	Inhibition/modulation/suppression of NLRP3 inflammasome	Apigenin, Catechin Quercetin Resveratrol Rutin	[86]
	Modulation/stimulation of AhR/Nrf2 pathway	Catechin Luteolin	[86]
	Inhibition of the expression of inflammatory factors via the MAPK and NF-kb signaling pathways	Baicalin Cinnamtannin B1 Isophilippinolide A Lincomolide A Secosubamolide Wogonoside	[14]
	Suppression of the NF-κB p65 translocation and block of the phosphorylation of IKK and IκB	Linalool	[87]
	Reduction of mRNA or protein expression of pro-inflammatory cytokines (IL-1 β, IL-6, and TNF-α) Up-regulation of mRNA and protein expression of anti-inflammatory cytokines (IL-10)	Geraniol 1,8-Cineole Linalool Thymol Salicilin 1,2, decanediol Salicylic acid Chlorogenic acid Caffeic acid Ferulic acid Vanillic acid Pedunculagin Cannabidiol Asiaticoside Asiatic acid Madecassic acid Silymarin Caffeine Saponin Cordytropolone Stipidalide	[21,27,34,38,39,42,88,89]
	Inhibition of COX and LOX activity	Geraniol 1,8-Cineole Linalool Thymol Caffeoylmalic acid 5-O-caffeoylquinic acid Chlorogenic acid Gentisic acid 2-O-glucoside Isoorinetin Isoquercitrin Quercitrin Hyperoside Rutin Swertiajaponin Scopoletin Linoleic acid Oleic acid *cis*-11-eicosenoic acid Palmitic acid Arachidic acid Palmitoleic acid Heneicosanoic acid Saponin Cordytropolone Stipidalide	[18,22,25,42,88,89]
	Inhibition of inducible nitric oxide synthase (iNOS) and tyrosinase expression	1,8-Cineole Geraniol Linalool Thymol	[88]

## Data Availability

Not applicable.

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
