# Peer review of "Bioactive Compounds from Medicinal Plants as Potential Adjuvants in the Treatment of Mild Acne Vulgaris"

_molecules, 2024, doi:10.3390/molecules29102394_

Round 1
Reviewer 1 Report
Comments and Suggestions for Authors
In the review entitled "Bioactive compounds from natural sources as potential adjuvants in the treatment of mild acne vulgaris" the author has described the source of natural polyhydroxy compounds like flavonoids, carbohydrates, terpenoids and their use in AV. 19 aceae families and the natural substance from them are described here. The way of presentation is nice. The only issue I want to point out here is the resolution of Figure1. It needs to improve.
I recommended to accept this article after minor correction.
Author Response
Reviewer #1
Comments and Suggestions for Authors
In the review entitled "Bioactive compounds from natural sources as potential adjuvants in the treatment of mild acne vulgaris" the author has described the source of natural polyhydroxy compounds like flavonoids, carbohydrates, terpenoids and their use in AV. 19 aceae families and the natural substance from them are described here. The way of presentation is nice. The only issue I want to point out here is the resolution of Figure1. It needs to improve.
I recommended to accept this article after minor correction.
- We sincerely thank the Reviewer for appreciating our work and for his efforts. Figure 1 has been deeply revised/resolved as recommended.
Reviewer 2 Report
Comments and Suggestions for Authors
The subsection 2. must be renamed 'Methodology' instead of 'Results'.
The full-form of abbreviations should be given in the first instance when they appear in the text, MIC, MBC etc.
Correct the spelling of 'three' which is given as 'tree' when quoting the databases sources of research under section 2.
Clarify if the course of action is topical application or ingestion for each source
Author Response
Reviewer #2
Comments and Suggestions for Authors
- We thank the Reviewer for his time and suggestions.
The subsection 2. must be renamed 'Methodology' instead of 'Results'.
- Subsection 2 renamed accordingly.
The full-form of abbreviations should be given in the first instance when they appear in the text, MIC, MBC etc.
- Revised accordingly.
Correct the spelling of 'three' which is given as 'tree' when quoting the databases sources of research under section 2.
- Typing error corrected. Thanks for noticing.
Clarify if the course of action is topical application or ingestion for each source
- To satisfy your request we have tried, where possible, to be more specific in terms of the type of applications. However, in many cases this can clearly be deduced from the studies carried out by the various authors. These are almost always topical applications considering both the type of pathology and the problems inherent to the well-known limited bioavailability of the plant-derived active ingredients for other routes of administration. In addition, many studies deal only with in vitro cell-based studies or even just theoretical approaches to identified a potential molecular target. Where the formulation (and its route of administration) has been prepared and in vivo or ex vivo studies have been carried out, we have reported and discussed the results.
Reviewer 3 Report
Comments and Suggestions for Authors
Mariateresa et al submitted the manuscript entitled: Bioactive compounds from natural sources as potential adjuvants in the treatment of mild acne vulgaris, in which they summarized recent advances on natural products for the treatment of AV. Generally, this is an informative manuscript and this topic would be of interest to potential readers of Molecule.
Here I list out some comments.
1. Figure 2 and other similar figures: For more instructive purposes, it is suggested to attach the corresponding biological data (for example, cytokine secretion and phenotype change, if applicable) into each figure. Most natural products can interact with multiple targets and the authors might be confused by which target(s) functioned in AV treatment. If multiple pathways are involved in AV therapy for a compound, a table may be a better way to list them.
2. Figure 1, while I can understand the current version of this figure, it seemed a bit messy. I suggest to move “C.acnes” to middle area, and extend the other content to 4 directions.
3. Figure 9, citation “De Tollenaere M. et al. (2022)”: I’m wondering the last sentence in this paragraph “Considering that the lipase activity … preventive and therapeutic treatment of AV” is a deduction from the authors or a proved result in this citation?
Author Response
Reviewer #3
Comments and Suggestions for Authors
Mariateresa et al submitted the manuscript entitled: Bioactive compounds from natural sources as potential adjuvants in the treatment of mild acne vulgaris, in which they summarized recent advances on natural products for the treatment of AV. Generally, this is an informative manuscript and this topic would be of interest to potential readers of Molecule.
- Thank you for appreciating our review article and for the overall positive feedback.
Here I list out some comments.
- Figure 2 and other similar figures: For more instructive purposes, it is suggested to attach the corresponding biological data (for example, cytokine secretion and phenotype change, if applicable) into each figure. Most natural products can interact with multiple targets and the authors might be confused by which target(s) functioned in AV treatment. If multiple pathways are involved in AV therapy for a compound, a table may be a better way to list them.
- Thank you for this suggestion. We thought a lot about how to best satisfy this request. The best thing we came up with (so as not to mix up all figures with data and information that are often not reliably comparable due to diversity of the experiments carried out) was to add a summary Table (i.e. Table 2) in which we reported all the proven mechanisms of action of the bioactive compounds we presented in the review article. We hope this is OK for you. After all, particularly significant data are already reported and discussed in the main text.
- Figure 1, while I can understand the current version of this figure, it seemed a bit messy. I suggest to move “C. acnes” to middle area, and extend the other content to 4 directions.
- Yes, we agree with this observation. The entire Figure 1 has been deeply revised accordingly, also in view of other Reviewers’ requests.
- Figure 9, citation “De Tollenaere M. et al. (2022)”: I’m wondering the last sentence in this paragraph “Considering that the lipase activity … preventive and therapeutic treatment of AV” is a deduction from the authors or a proved result in this citation?
- I guess you meant Figure 7, citation “De Tollenaere M. et al. (2022)”. However, yes you are right. This is actually a sentence from the conclusions of the original article that we forgot to remove from our draft. Sorry about that, and thanks for noticing. Of course, we have now deleted the sentence.